# Scavenging of reactive dicarbonyls with 2-hydroxybenzylamine reduces atherosclerosis in hypercholesterolemic *Ldlr*−/− mice

Huan Tao[1,5], Jiansheng Huang [1,5], Patricia G. Yancey [1], Valery Yermalitsky[2], John L. Blakemore[1], Youmin Zhang[1], Lei Ding[1], Irene Zagol-Ikapitte[3], Fei Ye[4], Venkataraman Amarnath[2], Olivier Boutaud[2], John A. Oates[2,3], L. Jackson Roberts II[2], Sean S. Davies[2] & MacRae F. Linton [1,2 ✉]

Lipid peroxidation generates reactive dicarbonyls including isolevuglandins (IsoLGs) and malondialdehyde (MDA) that covalently modify proteins. Humans with familial hypercholesterolemia (FH) have increased lipoprotein dicarbonyl adducts and dysfunctional HDL. We investigate the impact of the dicarbonyl scavenger, 2-hydroxybenzylamine (2-HOBA) on HDL function and atherosclerosis in *Ldlr*−/− mice, a model of FH. Compared to hypercholesterolemic *Ldlr*−/− mice treated with vehicle or 4-HOBA, a nonreactive analogue, 2-HOBA decreases atherosclerosis by 60% in *en face* aortas, without changing plasma cholesterol. *Ldlr*−/− mice treated with 2-HOBA have reduced MDA-LDL and MDA-HDL levels, and their HDL display increased capacity to reduce macrophage cholesterol. Importantly, 2-HOBA reduces the MDA- and IsoLG-lysyl content in atherosclerotic aortas versus 4-HOBA. Furthermore, 2-HOBA reduces inflammation and plaque apoptotic cells and promotes efferocytosis and features of stable plaques. Dicarbonyl scavenging with 2-HOBA has multiple atheroprotective effects in a murine FH model, supporting its potential as a therapeutic approach for atherosclerotic cardiovascular disease.

[1] Department of Medicine, Division of Cardiovascular Medicine, Atherosclerosis Research Unit, Vanderbilt University School of Medicine, Nashville, TN 37232, USA. [2] Department of Pharmacology, Vanderbilt University School of Medicine, Nashville, TN 37232, USA. [3] Department of Medicine, Division of Clinical Pharmacology, Vanderbilt University School of Medicine, Nashville, TN 37232, USA. [4] Department of Biostatistics, Vanderbilt University School of Medicine, Nashville, TN 37232, USA. [5] These authors contributed equally: Huan Tao, Jiansheng Huang. ✉email: macrae.linton@vanderbilt.edu

Atherosclerosis, the underlying cause of heart attack and stroke, is the most common cause of death and disability in the industrial world[1]. Elevated levels of apolipoprotein B (LDL and VLDL) containing lipoproteins and low levels of HDL increase the risk of atherosclerosis[1]. Although lowering LDL with HMG-CoA reductase inhibitors has been shown to reduce the risk of heart attack and stroke in large outcomes trials, substantial residual risk for cardiovascular events remains[2]. Atherosclerosis is a chronic inflammatory disease with oxidative stress playing a critical role[3,4]. Oxidative modification of apoB containing lipoproteins enhances internalization leading to foam cell

formation[1,5]. In addition, oxidized LDL induces inflammation, immune cell activation, and cellular toxicity[1,5]. HDL protects against atherosclerosis via multiple roles including promoting cholesterol efflux, preventing LDL oxidation, maintaining endothelial barrier function, and by minimizing cellular oxidative stress and inflammation[1,4,6]. HDL-C concentration is inversely associated with cardiovascular disease (CVD)[6], but recent studies suggest that assays of HDL function may provide new independent markers for CVD risk[7,8]. Evidence has mounted that oxidative modification of HDL compromises its functions, and studies suggest that oxidized HDL is indeed proatherogenic[1,6,9].

During lipid peroxidation, highly reactive dicarbonyls, including 4-oxo-nonenal (4-ONE), malondialdehyde (MDA), and isolevuglandins (IsoLGs) are formed. These reactive lipid dicarbonyls covalently bind to DNA, proteins, and phospholipid causing alterations in lipoprotein and cellular functions[1,10,11]. In particular, modification with reactive lipid dicarbonyls promotes inflammatory responses and toxicity that may be relevant to atherosclerosis[12–15]. Identifying effective strategies to assess the contribution of reactive lipid dicarbonyls to disease processes in vivo has been challenging. Although the formation of reactive lipid species, including dicarbonyls, theoretically could be suppressed simply by lowering levels of reactive oxygen species (ROS) using dietary antioxidants, the use of antioxidants to prevent atherosclerotic cardiovascular events has proven problematic with most clinical outcomes trials failing to show a benefit[1,16]. Dietary antioxidants like vitamin C and vitamin E are relatively ineffective suppressors of oxidative injury and lipid peroxidation. In fact, careful studies of patients with hypercholesterolemia found that the doses of vitamin E required to significantly reduce lipid peroxidation were substantially greater than those typically used in most clinical trials[17]. Furthermore, the high doses of antioxidants needed to suppress lipid peroxidation have been associated with significant adverse effects, likely because ROS play critical roles in normal physiology, including protection against bacterial infection and in a number of cell signaling pathways. Finally, for discovery purposes, the use of antioxidants provides little information about the role of reactive lipid dicarbonyls, because suppression of ROS inhibits formation of a broad spectrum of oxidatively modified macromolecules in addition to reactive lipid dicarbonyl species.

An alternative approach to broad suppression of ROS utilizing antioxidants is to use small molecule scavengers that selectively react with lipid dicarbonyl species without altering ROS levels, thereby preventing reactive lipid dicarbonyls from modifying cellular macromolecules without disrupting normal ROS signaling and function. 2-hydroxybenzylamine (2-HOBA) rapidly reacts with lipid dicarbonyls such as IsoLG, ONE, and MDA, but not with lipid monocarbonyls such as 4-hydroxynonenal[15,18–20]. The 2-HOBA isomer 4-hydroxybenzylamine (4-HOBA) is ineffective as a dicarbonyl scavenger[21]. Both of these compounds are orally bioavailable, so they can be used to examine the effects of lipid dicarbonyl scavenging in vivo[13,22]. 2-HOBA protects against oxidative stress associated hypertension[13], oxidant induced cytotoxicity[15], neurodegeneration,[14] and rapid pacing induced amyloid oligomer formation[23]. While there is evidence that reactive lipid dicarbonyls play a role in atherogenesis[6,7], to date the effects of scavenging lipid dicarbonyl on the development of atherosclerosis have not been examined.

Here, we show that 2-HOBA treatment significantly attenuates atherosclerosis development in hypercholesterolemic *Ldlr*$^{-/-}$ mice. Importantly, 2-HOBA treatment inhibits cell death and necrotic core formation in lesions, leading to the formation of characteristics of more stable plaques as evidenced by increased lesion collagen content and fibrous cap thickness. Consistent with the decrease in atherosclerosis from 2-HOBA treatment being

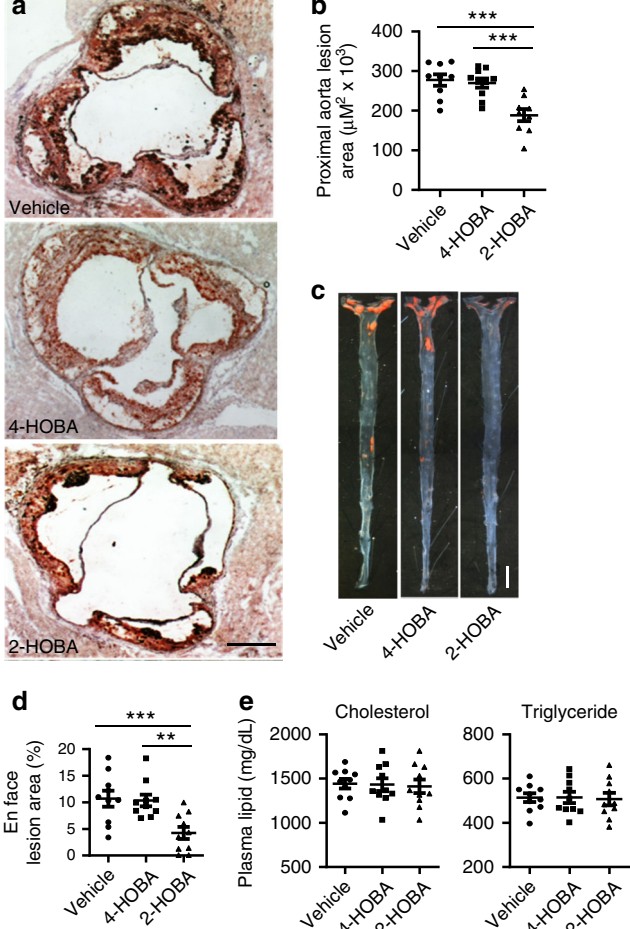

**Fig. 1 2-HOBA attenuates atherosclerosis in hypercholesterolemic female** ***Ldlr*$^{-/-}$ mice.** 8-week-old *Ldlr*$^{-/-}$ female mice were pretreated with 1 g/L 2-HOBA or 1 g/L 4-HOBA (nonreactive analog) or vehicle (water) for 2 weeks and then the treatment was continued for 16 weeks during which the mice were fed a Western diet. Representative images (**a**) and quantitation (**b**) of Oil-Red-O stained proximal aorta root sections. Scale bar = 500 μm. $n = 9$ (vehicle), 10 (4-HOBA), or 9 (2-HOBA) biologically independent mice. Data are presented as mean ± SEM, One-way ANOVA with Bonferroni's post hoc test, *p* values of 2-HOBA vs 4-HOBA and 2-HOBA vs vehicle are ***0.0009 and ***0.0004. Representative images (**c**) and quantitation (**d**) of Oil-Red-O stained open-pinned aortas. $n = 10$ biologically independent mice per group. Data are presented as mean ± SEM. One-way ANOVA with Bonferroni's post hoc test, *p* values of 2-HOBA vs 4-HOBA and 2-HOBA vs vehicle are **0.0055 and **0.0033. **e** The plasma total cholesterol and triglyceride levels. $n = 10$ biologically independent mice per group. Data are presented as mean ± SEM. One-way ANOVA with Bonferroni's post hoc test, *p* values of 2-HOBA vs 4-HOBA and 2-HOBA vs vehicle are >0.9999 for both cholesterol and triglyceride levels. Source data are provided as a source data file.

due to scavenging of reactive dicarbonyls, the atherosclerotic lesion MDA and IsoLG adduct content was markedly reduced in 2-HOBA treated vs control mice. We further show that 2-HOBA treatment results in decreased MDA-LDL and MDA-HDL. In addition, MDA-apoAI adduct formation was decreased, and importantly, 2-HOBA treatment caused more efficient HDL function in reducing macrophage cholesterol stores. Thus, scavenging of reactive carbonyls with 2-HOBA has multiple antiatherogenic therapeutic effects that likely contribute to its ability to reduce the development of atherosclerosis in hypercholesterolemic $Ldlr^{-/-}$ mice. We also found that HDL from humans with severe familial hypercholesterolemia (FH) contained increased MDA adducts vs control subjects, and that FH-HDL were extremely impaired in reducing macrophage cholesterol stores. Taken together, our studies raise the possibility that reactive dicarbonyl scavenging represents a therapeutic approach for atherosclerotic CVD in humans.

## Results

**2-HOBA treatment reduces atherosclerosis in $Ldlr^{-/-}$ mice.** Eight-week-old female $Ldlr^{-/-}$ mice were fed a western-type diet for 16 weeks and were continuously treated with vehicle alone (water) or water containing either 2-HOBA or 4-HOBA, an ineffective dicarbonyl scavenger. Treatment with 2-HOBA reduced the extent of proximal aortic atherosclerosis by 31.1% and 31.5%, compared to treatment with either vehicle or 4-HOBA, respectively (Fig. 1a, b). In addition, en face analysis of the aorta demonstrated that treatment of female $Ldlr^{-/-}$ mice with 2-HOBA reduced the extent of aortic atherosclerosis by 60.3% and 59.1% compared to administration of vehicle and 4-HOBA, respectively (Fig. 1c, d). Compared to administration of vehicle or 4-HOBA, 2-HOBA treatment did not affect body weight, water consumption, or diet uptake (Supplementary Fig. 1A–C). In addition, the plasma total cholesterol and triglyceride levels were not significantly different (Fig. 1e), and the lipoprotein distribution was similar between the three groups of mice (Supplementary Fig. 1D). Consistent with these results, the treatment of male $Ldlr^{-/-}$ mice with 1 g/L of 2-HOBA, under similar conditions of being fed a western diet for 16 weeks, reduced the extent of proximal aortic and whole aorta atherosclerosis by 37% and 45%, respectively, compared to treatment with water (Supplementary Fig. 2A–D) but did not affect the plasma total cholesterol levels (Supplementary Fig. 2E). Thus, we demonstrate that 2-HOBA treatment significantly decreases atherosclerosis development in an experimental mouse model of FH without changing plasma cholesterol and triglyceride levels. Examination of the proximal aortic MDA adduct content by immunofluorescence staining using an antibody against MDA-protein adducts (Abcam cat# ab6463) shows that the MDA adduct levels were reduced by 68.5 and 66.8% in 2-HOBA treated mice compared to mice treated with vehicle alone or 4-HOBA (Fig. 2a, b). We determined that the anti-MDA-protein antibody does not recognize either free MDA or MDA-2-HOBA adducts (Supplementary Fig. 3A, B). In addition, 2-HOBA does not interfere with the antibody recognition of MDA-albumin adducts (Supplementary Fig. 3A, B). Quantitative measurement of the whole aorta MDA- and IsoLG-lysyl adducts by LC/MS/MS demonstrates that compared to 4-HOBA treatment, administration of 2-HOBA decreased the MDA and IsoLG adduct content by 59% and 23%, respectively (Fig. 2c, d). We determined by LC/MS/MS that the plasma levels of 2-HOBA in the male $Ldlr^{-/-}$ mice after 16 weeks of treatment with 1 g of 2-HOBA/L of water were $469 \pm 38$ ng/mL, which is similar to what we previously reported in C57BL6 mice receiving 1 g/L of 2-HOBA[22]. In addition, these levels are in the same range as the plasma

2-HOBA levels in humans in a recent safety trial[24]. The plasma levels of 4-HOBA in the male $Ldlr^{-/-}$ mice after 16 weeks of treatment with 1 g of 4-HOBA/L of water were $25 \pm 3$ ng/mL. However, the plasma levels of 2-HOBA vs 4-HOBA in male $Ldlr^{-/-}$ mice 30 min after oral gavage of 5 mg were not significantly different (Supplementary Fig. 4A). In addition, the levels of 2-HOBA and 4-HOBA were similar in the aorta and heart of male $Ldlr^{-/-}$ mice 30 min after oral gavage (Supplementary Fig. 4B, C). While plasma levels of 4-HOBA after intraperitoneal injection were slightly higher initially than those of 2-HOBA, 4-HOBA appeared to undergo more rapid clearance (Supplementary Fig. 5A). In addition, the liver, spleen, and kidney levels of 2-HOBA vs 4-HOBA were not significantly different 30 min after intraperitoneal injection (Supplementary Fig. 5B, D). Taken together, the lower levels of 4-HOBA vs 2-HOBA in the male $Ldlr^{-/-}$ mice after 16 weeks of treatment are likely due to differences in clearance as well as in timing of water consumption before sacrifice. Interestingly, the IsoLG-2-HOBA adducts (with masses consistent with potential keto-pyrrole, anhydro-lactam, keto-lactam, pyrrole, and anhydro-hydroxylactam adducts) were present in the hearts and livers of $Ldlr^{-/-}$ mice after 16 weeks on a western diet, whereas IsoLG-4-HOBA adducts were non-detectable (Supplementary Fig. 6 and Supplementary Table 1). Importantly, the MDA-2-HOBA vs MDA-4-HOBA adducts (with mass consistent with propenal-HOBA adducts) were increased by 19-fold in the urine collected during 16 h after oral gavage (5 mg) treatment of $Ldlr^{-/-}$ mice fed a western diet for 16 weeks (Supplementary Fig. 7A). In addition, the liver, kidney, and spleen from 2-HOBA vs 4-HOBA treated $Ldlr^{-/-}$ mice also contained 3-, 5-, and 11-fold more propenal-HOBA adducts 16 h post oral gavage (Supplementary Fig. 7B, D). Urine F2-isoprostane (IsoP) levels are a measure of systemic lipid peroxidation, and treatment of $Apoe^{-/-}$ mice with the antioxidant alpha tocopherol reduces atherosclerosis and urine F2-IsoP levels[25,26]. Importantly, we found that the urine F2-IsoP levels were not different in $Ldlr^{-/-}$ mice treated with vehicle, 4-HOBA, and 2-HOBA (Supplementary Fig. 8), indicating that the effects of 2-HOBA on atherosclerosis were not due to general inhibition of lipid peroxidation or metal ion chelation. Taken together these results support the hypothesis that the impact of 2-HOBA on atherosclerosis is due to reactive lipid dicarbonyl scavenging.

**2-HOBA promotes formation of features of stable plaques.** As vulnerable plaques exhibit higher risk for acute cardiovascular events in humans[1], we examined the effects of 2-HOBA treatment on characteristics of plaque stabilization by quantitating the atherosclerotic lesion collagen content, fibrous cap thickness, and necrotic cores (Fig. 3a–d). Compared to administration of vehicle or 4-HOBA, 2-HOBA treatment increased the collagen content of the proximal aorta by 2.7- and 2.6-fold, respectively (Fig. 3a, b). In addition, the fibrous cap thickness was 2.31- and 2.29-fold greater in lesions of 2-HOBA treated mice vs vehicle and 4-HOBA treated mice (Fig. 3a, c). Importantly, the % of necrotic area in the proximal aorta was decreased by 74.8% and 73.5% in mice treated with 2-HOBA vs vehicle and 4-HOBA (Fig. 3a, d). Taken together, these data show that 2-HOBA suppresses the characteristics of vulnerable plaque formation in the hypercholesterolemic $Ldlr^{-/-}$ mice.

**2-HOBA promotes efferocytosis and reduces inflammation.** As enhanced cell death and insufficient efferocytosis promote necrotic core formation and destabilization of atherosclerotic plaques, we next examined the effects of 2-HOBA treatment on cell death and efferocytosis in atherosclerotic lesions in the proximal aorta (Fig. 4a–d). Compared to treatment with either

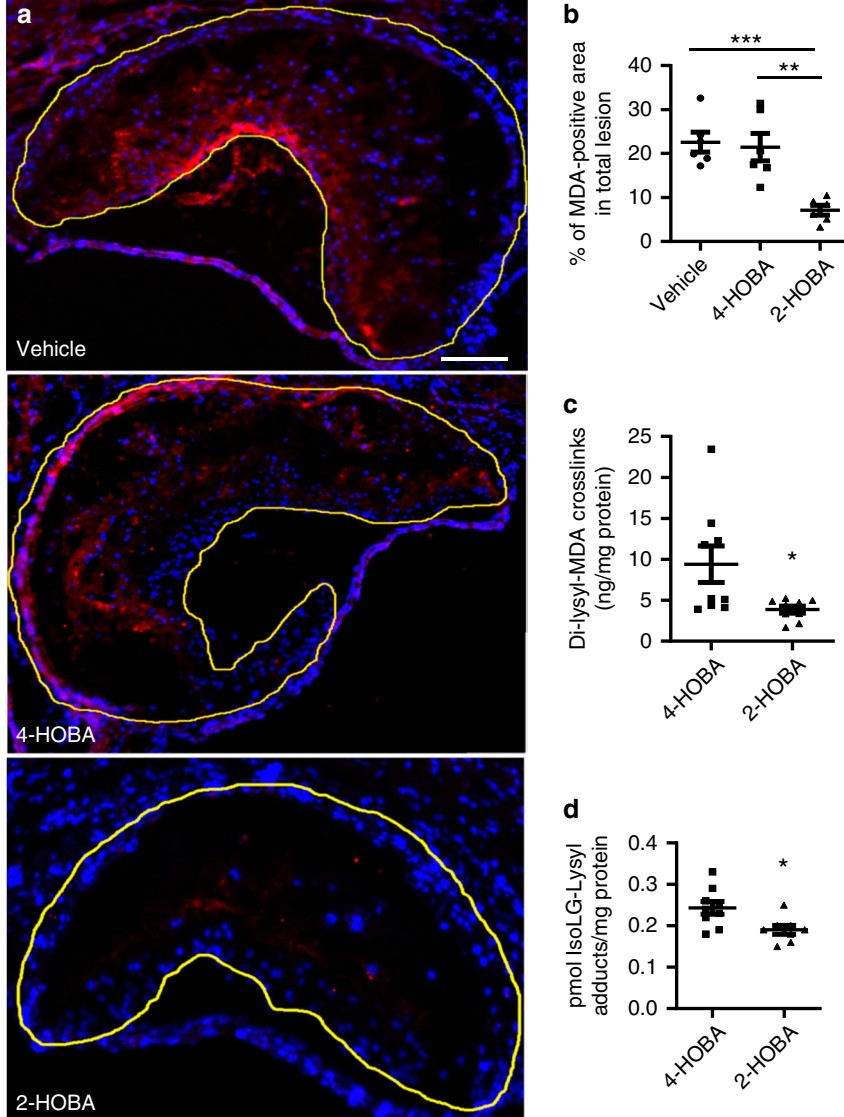

**Fig. 2 2-HOBA decreases the MDA adduct content of proximal aortic atherosclerotic lesions in $Ldlr^{-/-}$ mice. a, b** MDA was detected by immunofluorescence using anti-MDA primary antibody and fluorescent-labeled secondary antibody (red). Nuclei were counterstained with Hoechst (blue). Representative images (**a**) and quantitation (**b**) of MDA staining in proximal aortic root sections. Scale Bar = 50 μm. $n = 6$ biologically independent mice per group. Data are expressed as mean ± SEM. One-way ANOVA with Bonferroni's post hoc test, $p$ values of 2-HOBA vs 4-HOBA and 2-HOBA vs vehicle are **0.0017 and ***0.0008, respectively. **c** Aortic tissues were isolated from the $Ldlr^{-/-}$ mice and Dilysyl-MDA crosslinks were measured by LC/MS/MS. $n = 9$ biologically independent mice per group. Data are presented as mean ± SEM. Two-sided unpaired $t$ test, $p$ value of 2-HOBA vs 4-HOBA is *0.0262. **d** Aortic tissues were isolated from the $Ldlr^{-/-}$ mice and IsoLG-Lysyl was measured by LC/MS/MS. $n = 9$ (4-HOBA) or 8 (2-HOBA) biologically independent mice. Data are presented as mean ± SEM. Two-sided unpaired $t$ test, $p$ value of 2-HOBA vs 4-HOBA is *0.0151. Source data are provided as a source data file.

vehicle or 4-HOBA, the number of TUNEL-positive cells was reduced by 72.9 and 72.4% in the proximal aortic lesions of 2-HOBA treated mice (Fig. 4a–c). Next, we examined the impact of 2-HOBA on efferocytosis in the atherosclerotic lesions, and the number of TUNEL-positive cells not associated with macrophages was increased by 1.9- and 2.0-fold in lesions of mice treated with vehicle and 4-HOBA vs 2-HOBA (Fig. 4b, d), supporting the ability of reactive lipid dicarbonyl scavenging to maintain efficient efferocytosis. Consistent with lesion necrosis being linked to enhanced inflammation, the serum levels of IL-1β, IL-6, TNF-α, and serum amyloid A were reduced in 2-HOBA vs 4-HOBA or vehicle treated $Ldlr^{-/-}$ mice (Fig. 5), suggesting that reactive dicarbonyl scavenging decreased systemic inflammation. In contrast to our results in $Ldlr^{-/-}$ mice fed a western diet,

$Ldlr^{-/-}$ mice consuming a chow diet had lower plasma levels of IL-1β, IL-6, and TNF-α, and 2-HOBA treatment had no impact on cytokine levels in the chow fed mice (Supplementary Fig. 9). These results support the ability of a high-fat western diet to induce oxidative stress and inflammation in $Ldlr^{-/-}$ mice. As studies have demonstrated that incubation of cells with either $H_2O_2$[27,28] or oxidized LDL[29–31] induces lipid peroxidation, inflammation, and death, we next determined the in vitro effects of reactive dicarbonyl scavenging on the cellular response to oxidative stress. Examination of the susceptibility of macrophages and endothelial cells to apoptosis in response to $H_2O_2$ treatment demonstrates that compared to incubation with vehicle or 4-HOBA, 2-HOBA markedly decreased the number of apoptotic cells in both macrophage and endothelial cell cultures (Fig. 6a, b).

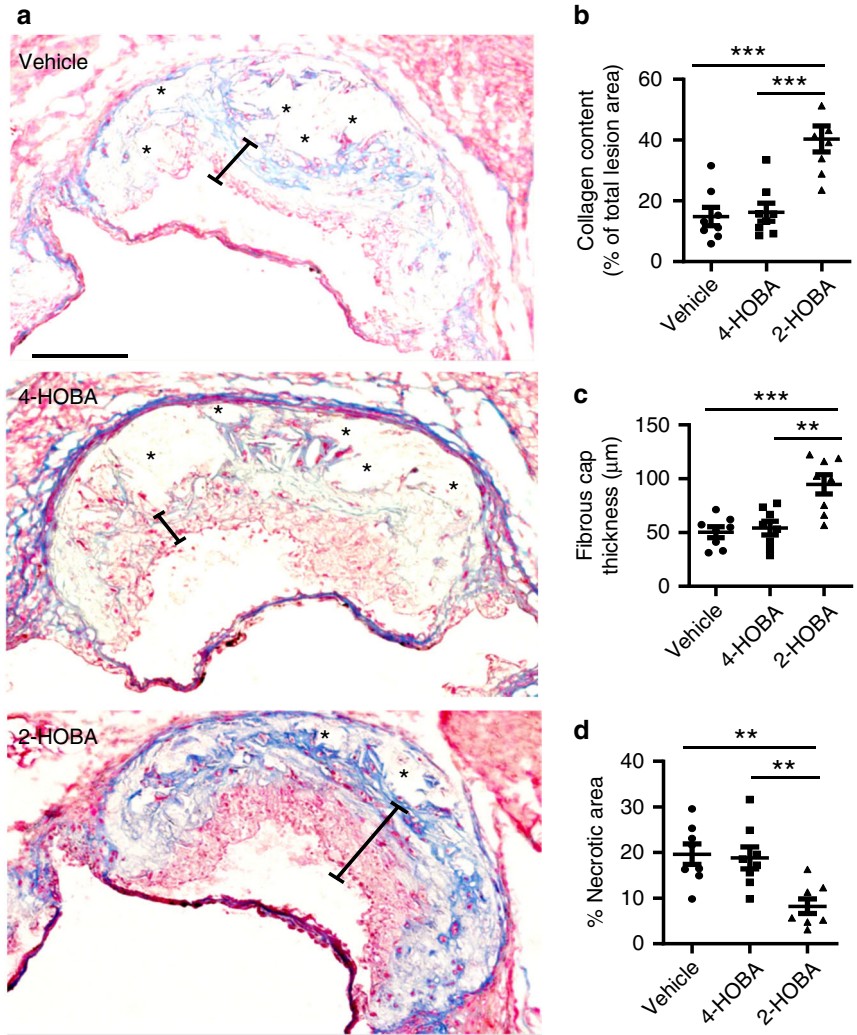

**Fig. 3 2-HOBA promotes features of stable atherosclerotic plaques in _Ldlr_$^{-/-}$ mice. a–d** Masson's Trichrome stain was done to analyze characteristics associated with atherosclerotic lesion stability in proximal aorta sections of _Ldlr_$^{-/-}$ mice. Representative images (**a**) of Masson's Trichrome stain in aorta root sections and quantitation of aortic root collagen content (**b**), fibrous cap thickness (**c**), and necrotic area (**d**). Blue shows collagen, red, cytoplasm, black, nuclei. Scale bar = 50 μm. Quantitations were done using ImageJ software. **a–d** n = 8 biologically independent mice per group. Data are presented as mean ± SEM. One-way ANOVA with Bonferroni's post hoc test. **b** p values of 2-HOBA vs 4-HOBA and 2-HOBA vs vehicle are ***0.0002 and ***0.0001. **c** p values of 2-HOBA vs 4-HOBA and 2-HOBA vs vehicle are **0.0015 and ***0.0006. **d** p values of 2-HOBA vs 4-HOBA and 2-HOBA vs vehicle are **0.0058 and **0.0031. Source data are provided as a source data file.

In addition, 2-HOBA treatment significantly reduced the macrophage inflammatory response to oxidized LDL as shown by the decreased mRNA levels of IL-1β, IL-6, and TNF-α (Fig. 6c–e). Similar results were observed for the impact of 2-HOBA on the inflammatory cytokine response of macrophages treated with $H_2O_2$ vs vehicle or 4-HOBA treatment (Fig. 6f–h). In addition, the levels of IL-1β, IL-6, and TNF-α mRNA were significantly reduced in macrophages treated with only 5 μM 2-HOBA (615 ng/mL) in the presence of $H_2O_2$ (Supplementary Fig. 10). Consistent with the 2-HOBA effects on cell death and inflammation being due to scavenging reactive dicarbonyls, the levels of MDA-2-HOBA (propenal-2-HOBA) vs MDA-4-HOBA adducts were increased in cells treated with oxidized LDL and 250 or 500 μM 2-HOBA (Supplementary Fig. 11A). Even in cells incubated with just 5 μM 2-HOBA, significant levels of propenal-2-HOBA were formed as well as DHP-MDA-2-HOBA, and crosslinked MDA-2-HOBA adducts were detected (Supplementary Fig. 11B, D). In addition, 2-HOBA did not have a direct effect on prosurvival, anti-inflammatory signaling in the absence of oxidative stress[32], as there was no difference in pAKT levels in macrophages treated

with vehicle, 4-HOBA, and 2-HOBA in the absence and presence of insulin (Supplementary Fig. 12). Due to the striking impact on inflammatory cytokines in vivo, we also measured urinary prostaglandins to evaluate whether 2-HOBA might be inhibiting cyclooxygenase (COX). Urine samples were analyzed for 2,3-dinor-6-keto-PGF1, 11-dehydro TxB2, PGE-M, PGD-M by LC/MS. We found that there were no significant differences in levels of these major urinary prostaglandin metabolites of _Ldlr_$^{-/-}$ mice treated with 2-HOBA compared to the vehicle control (Supplementary Fig. 13), indicating that 2-HOBA was not significantly inhibiting COX in vivo in mice. Taken together, these data show that 2-HOBA treatment maintains efficient efferocytosis in vivo and prevents apoptosis and inflammation in response to oxidative stress by scavenging reactive dicarbonyls.

**Impact of 2-HOBA on lipoprotein MDA content and function.** Treatment of the _Ldlr_$^{-/-}$ mice fed a western diet for 16 weeks with 2-HOBA vs 4-HOBA or vehicle decreased the plasma levels of MDA (Supplementary Fig. 14A). Compared to treatment with either vehicle or 4-HOBA, the MDA adduct content in isolated

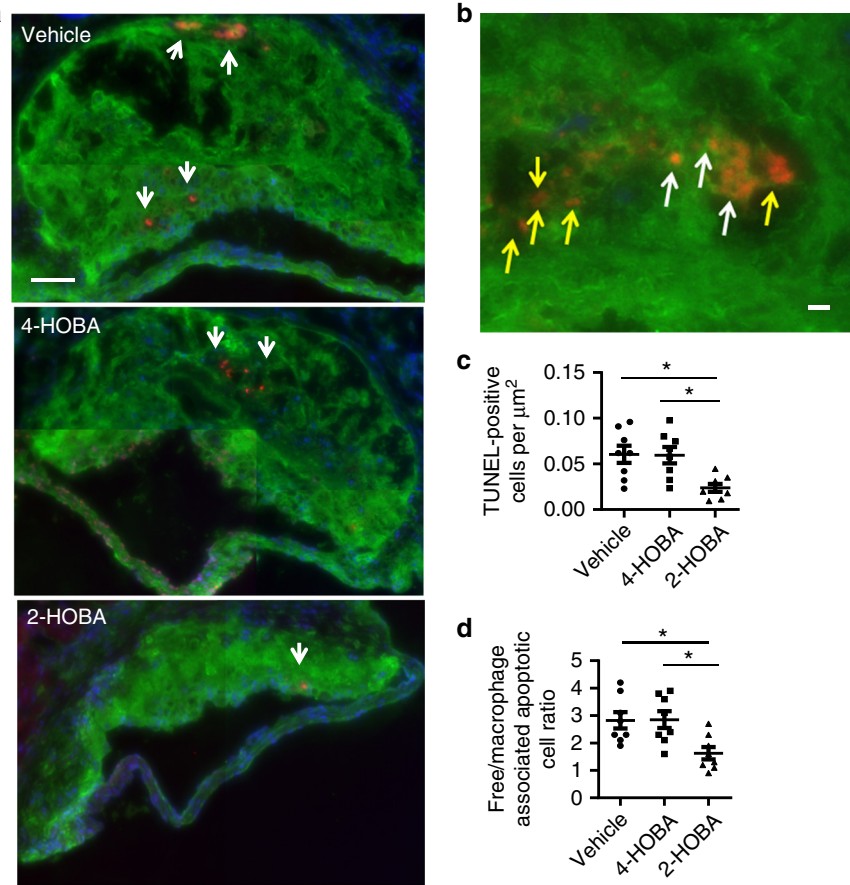

**Fig. 4 2-HOBA prevents cell death and increases efferocytosis in atherosclerotic lesions of *Ldlr*⁻/⁻ mice. a**, **c** Representative images **a** and quantitation **c** of TUNEL-positive nuclei (red) of proximal aorta sections. Macrophages were detected by anti-macrophage primary antibody (green), and nuclei were counterstained with Hoechst (blue). Scale bar = 50 μm. $n = 8$ biologically independent mice per group. Data are presented as mean ± SEM. One-way ANOVA with Bonferroni's post hoc test, $p$ values of 2-HOBA vs 4-HOBA and 2-HOBA vs vehicle are *0.0134 and *0.0108. Higher magnification images of the TUNEL staining (**b**) were used to quantitate (**d**) efferocytosis of dead cells in aortic root sections. A representative image taken at a higher magnification shows macrophage-associated TUNEL stain (white arrows) and free dead cells (yellow arrows) that were not associated with macrophages (**b**). Scale bar = 50 μm. Efferocytosis was quantitated as the free vs macrophage-associated TUNEL-positive cells in the proximal aortic sections (**d**). $n = 8$ biologically independent mice per group. Data are presented as mean ± SEM. One-way ANOVA with Bonferroni's post hoc test, $p$ values of 2-HOBA vs 4-HOBA and 2-HOBA vs vehicle are *0.0162 and *0.0187. Source data are provided as a source data file.

LDL measured by ELISA was reduced by 57% and 54%, respectively, in *Ldlr*⁻/⁻ mice treated with 2-HOBA (Supplementary Fig. 14B). By comparison, LDL from control and FH subjects contained similar amounts of MDA adducts, which were not significantly different (Supplementary Fig. 14C). MDA modification of LDL induces foam cell formation and examination of the ability of LDL from 2-HOBA vs 4-HOBA or vehicle treated *Ldlr*⁻/⁻ mice to enrich cells with cholesterol was not different (Supplementary Fig. 14D). This observation was due to the plasma LDL from hypercholesterolemic *Ldlr*⁻/⁻ mice being insufficiently modified with MDA to induce cholesterol loading as we determined by in vitro modification of LDL that the MDA content must be 2500 ng/mg LDL protein to enrich cells with cholesterol. As oxidative modification of HDL impairs its functions, we next examined the effects of 2-HOBA treatment on HDL MDA content and function. Treatment of *Ldlr*⁻/⁻ mice with 2-HOBA reduced the MDA adduct content of isolated HDL as measured by ELISA by 57% and 56% (Fig. 7a) compared to treatment with either vehicle or 4-HOBA. Next, we examined the effects of 2-HOBA on apoAI MDA adduct formation. ApoAI was isolated from plasma by immunoprecipitation, and MDA-apoAI was detected by western blotting with the antibody to MDA-protein adducts. After 16 weeks on the western-type diet, *Ldlr*⁻/⁻

mice treated with vehicle or 4-HOBA had markedly increased plasma levels of MDA-apoAI compared to *Ldlr*⁻/⁻ mice consuming a chow diet (Fig. 7b, c). In contrast, treatment of *Ldlr*⁻/⁻ mice consuming a western diet with 2-HOBA dramatically reduced plasma MDA-apoAI adducts (Fig. 7b, c). The levels of apoAI were similar among the four groups of mice (Fig. 7b). Importantly, the HDL isolated from 2-HOBA treated *Ldlr*⁻/⁻ mice was 2.2- and 1.7-fold more efficient at reducing cholesterol stores in *Apoe*⁻/⁻ macrophage foam cells vs vehicle and 4-HOBA treated mice (Fig. 7d). In addition, HDL from human subjects with severe FH pre- and post-LDL apheresis (LA) had 5.9-fold and 5.6-fold more MDA adducts compared to control HDL as measured by ELISA (Fig. 7e). We also found that the dilysyl-MDA crosslink levels as measured by LC/MS/MS were higher in HDL from FH vs control subjects (Fig. 7f). Importantly, HDL from FH vs control subjects lacked the ability to reduce the cholesterol content of cholesterol-enriched *Apoe*⁻/⁻ macrophages (Fig. 7g). While the effects of MDA modification of lipid-free apoAI on cholesterol efflux are established[33], studies are controversial regarding the impact of modification of HDL[34,35]. Therefore, we determined the impact of in vitro modification of HDL with MDA on the the ability of HDL to reduce the cholesterol content of macrophage foam cells as it relates to the MDA

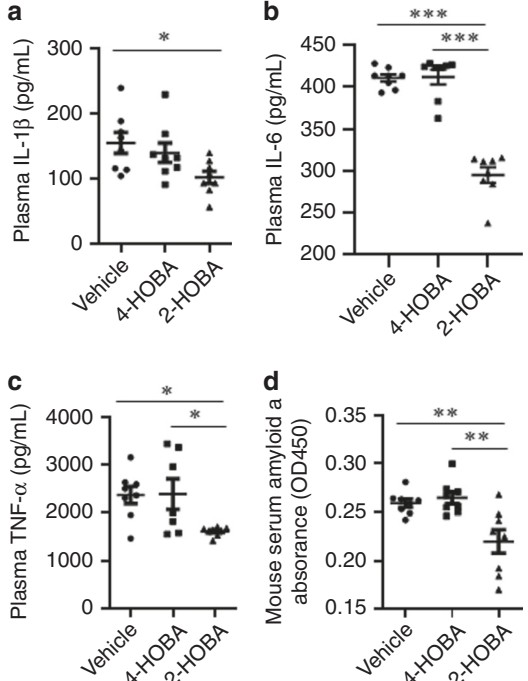

**Fig. 5 2-HOBA reduces plasma inflammatory cytokines in hypercholesterolemic Ldlr⁻/⁻ mice. a–d** Inflammatory cytokines were measured in plasma from mice consuming a western diet for 16 weeks and treated with 2-HOBA, 4-HOBA, or vehicle. **a** IL-1β levels were measured by ELISA. $n = 8$ biologically independent mice per group. Data are presented as mean ± SEM. One-way ANOVA with Bonferroni's post hoc test, $p$ value of 2-HOBA vs vehicle is *0.0401 and 2-HOBA vs 4-HOBA is not significant. **b** IL-6 levels were measured by ELISA. $n = 8$ biologically independent mice per group. Data are presented as mean ± SEM. One-way ANOVA with Bonferroni's post hoc test, $p$ values of 2-HOBA vs 4-HOBA and 2-HOBA vs vehicle are ***<0.0001. **c** TNF-α levels were measured by ELISA. $n = 8$ (vehicle), 7 (4-HOBA), or 8 (2-HOBA) biologically independent mice. Data are presented as mean ± SEM. One-way ANOVA with Bonferroni's post hoc test, $p$ values of 2-HOBA vs 4-HOBA and 2-HOBA vs vehicle are *0.0385 and *0.0359. **d** The levels of SAA were measured by ELISA. $n = 8$ biologically independent mice per group. Data are presented as mean ± SEM. One-way ANOVA with Bonferroni's post hoc test, $p$ values of 2-HOBA vs 4-HOBA and 2-HOBA vs vehicle are **0.0025 and **0.0082. Source data are provided as a source data file.

adduct content measured by ELISA (Supplementary Fig. 15A, B). MDA modification of HDL inhibited the net cholesterol efflux capacity in a dose dependent manner, and, importantly, the MDA-HDL adduct levels, which impacted the cholesterol efflux function, were in the same range as MDA adduct levels in HDL from FH subjects and hypercholesterolemic Ldlr⁻/⁻ mice. Taken together, dicarbonyl scavenging with 2-HOBA prevents macrophage foam cell formation by improving HDL net cholesterol efflux capacity. In addition, our studies suggest that scavenging of reactive lipid dicarbonyls could be a relevant therapeutic approach for improving HDL function in humans given that HDL from subjects with homozygous FH contain increased MDA and IsoLG and enhanced foam cell formation.

## Discussion

Oxidative stress-induced lipid peroxidation has been implicated in the development of atherosclerosis. Genetic defects and/or environmental factors cause an imbalance between oxidative stress and the ability of the body to counteract or detoxify the harmful effects of oxidation products[1,3,36]. The large body of

experimental evidence implicating an important role of lipid peroxidation in the pathogenesis of atherosclerosis previously had stimulated interest in the potential for antioxidants to prevent atherosclerotic CVD. Although a few trials of dietary antioxidants in humans demonstrated reductions in atherosclerosis and cardiovascular events, the majority of large clinical outcomes trials with antioxidants have failed to show any benefit in terms of reduced cardiovascular events. Possible reasons for the failure of these trials to reduce cardiovascular events include inadequate doses of antioxidants being used in the trials[1,16] and the inhibition of normal ROS signaling that may be antiatherogenic[37].

Peroxidation of lipids in tissues/cells or in blood produces a number of reactive lipid carbonyls and dicarbonyls including 4-hydroxynonenal, methylglyoxal, MDA, 4-ONE, and IsoLGs. These electrophiles can covalently bind to proteins, phospholipids, and DNA causing alterations in lipoprotein and cellular functions[1,10,11]. Treatment with scavengers of reactive lipid carbonyl and dicarbonyl species represents an alternative therapeutic strategy that will decrease the adverse effects of a particular class of bioactive lipids without completely inhibiting the normal signaling mediated by ROS[37]. A number of compounds with the potential to scavenge carbonyls have been identified, with individual compounds preferentially reacting with different classes of carbonyls so that the effectiveness of a scavenging compound in mitigating disease can serve as an indicator that their target class of carbonyl contributes to the disease process[37]. Previous studies found that scavengers of methylglyoxal and glyoxal, such as aminoguanidine and pyridoxamine, reduce atherosclerotic lesions in streptozotocin-treated Apoe⁻/⁻ mice[38,39]. Similarly, scavengers of α,β-unsaturated carbonyls (e.g., HNE and acrolein) such as carnosine and its derivatives, also reduce atherosclerosis in Apoe⁻/⁻ mice or streptozotocin-treated Apoe⁻/⁻ mice[40–42]. These previously tested scavenger compounds are poor in vivo scavengers of lipid dicarbonyls such as IsoLG and MDA[37]. Therefore, we sought to examine the potential of 2-HOBA, an effective scavenger of IsoLG and MDA, to prevent the development of atherosclerosis in Ldlr⁻/⁻ mice.

We have recently reported that 2-HOBA can reduce IsoLG-mediated HDL modification and dysfunction[43]. Our current studies are the first to examine the effects of dicarbonyl scavenging on atherosclerosis, and we demonstrate that the dicarbonyl scavenger, 2-HOBA, significantly reduces atherosclerosis development in the hypercholesterolemic Ldlr⁻/⁻ mouse model of FH (Fig. 1). Importantly, our studies show that 2-HOBA treatment markedly improves features of the stability of the atherosclerotic plaque as evidenced by decreased necrosis and increased fibrous cap thickness and collagen content (Fig. 3). Consistent with the proinflammatory effects of reactive dicarbonyls[43] and the impact on lesion necrosis, 2-HOBA reduced systemic inflammation by neutralizing reactive dicarbonyls (Figs. 5 and 6). Furthermore, dicarbonyl scavenging reduced in vivo MDA modification of HDL, consistent with the notion that preventing dicarbonyl modification of HDL improves its net cholesterol efflux capacity (Fig. 7). We previously showed that IsoLG modification increases in HDL from subjects with FH[43], and this current study shows that MDA modification is similarly increased (Fig. 7), suggesting these modifications contribute to the enhanced foam cell formation induced by FH-HDL (Fig. 7). Taken together, dicarbonyl scavenging using 2-HOBA offers therapeutic potential in reducing atherosclerosis development and the risk of clinical events resulting from formation of vulnerable atherosclerotic plaques.

As our studies show that 2-HOBA reduces atherosclerosis development without decreasing plasma cholesterol levels (Fig. 1), the atheroprotective effects of 2-HOBA are likely due to scavenging bioactive dicarbonyls. Consistent with this concept, the atherosclerotic lesion MDA- and IsoLG-lysyl adducts were

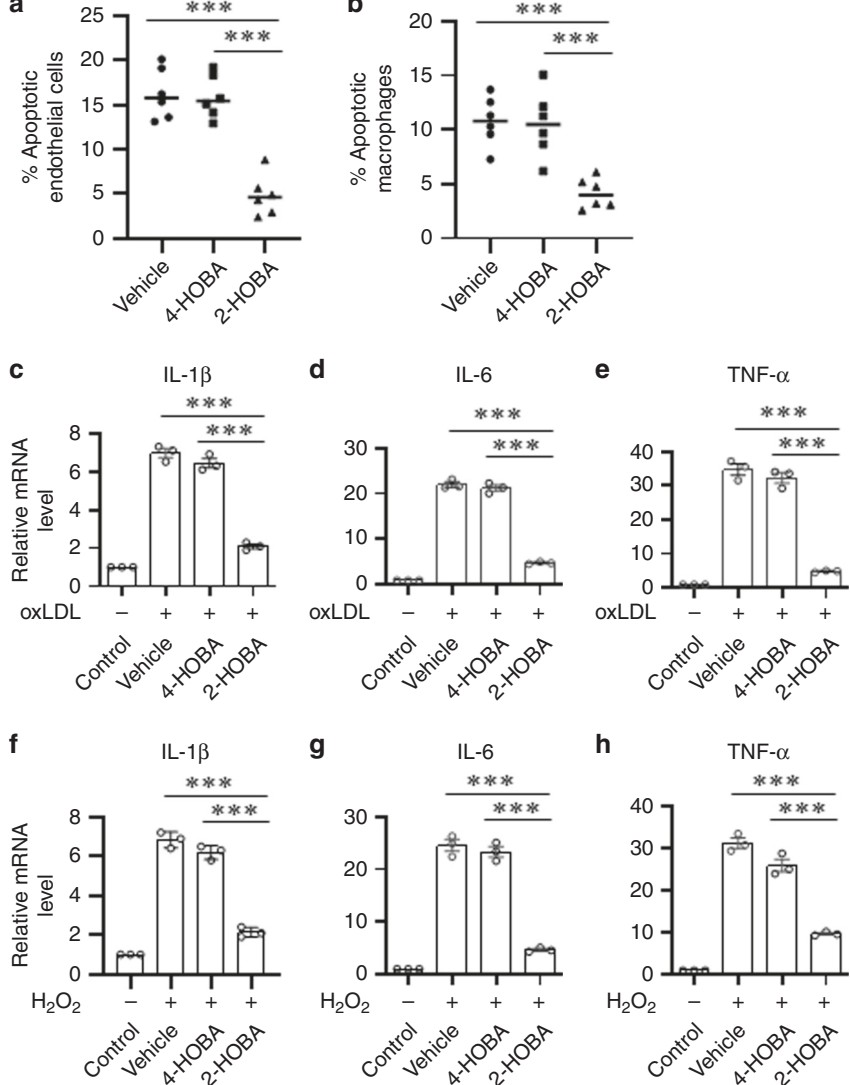

**Fig. 6 In vitro treatment with 2-HOBA suppresses oxidative stress-induced cell apoptosis and inflammation. a** Human aortic endothelial cells were incubated for 24 h with 250 µM $H_2O_2$ alone or with either 4-HOBA or 2-HOBA (500 µM). Apoptotic cells were then detected by Annexin V staining and flow cytometry. $n = 6$ biologically independent experiments per group. Data are presented as mean ± SEM. One-way ANOVA with Bonferroni's post hoc test, $p$ values of 2-HOBA vs 4-HOBA and 2-HOBA vs vehicle are ***<0.0001. **b** Mouse primary macrophages were incubated for 24 h with 250 µM $H_2O_2$ alone or with either 4-HOBA or 2-HOBA (500 µM). $n = 6$ biologically independent experiments per group. Data are presented as mean ± SEM. One-way ANOVA with Bonferroni's post hoc test, $p$ values of 2-HOBA vs 4-HOBA and 2-HOBA vs vehicle are ***0.0006 and ***0.0009. **c–e** The mRNA levels of IL-1β, IL-6, and TNF-α were analyzed by real-time PCR in the peritoneal macrophages incubated for 24 h with either oxidized LDL alone or with either 4-HOBA or 2-HOBA (500 µM). $n = 3$ biologically independent experiments per group. Data are presented as mean ± SEM. One-way ANOVA with Bonferroni's post hoc test, $p$ values of 2-HOBA vs vehicle and 2-HOBA vs 4-HOBA are ***<0.0001. **f–h** The mRNA levels of IL-1β, IL-6, and TNF-α were analyzed by real-time PCR in the peritoneal macrophages incubated for 24 h with either 250 µM $H_2O_2$ alone or with either 4-HOBA or 2-HOBA (500 µM). $n = 3$ biologically independent experiments per group. Data are presented as mean ± SEM. One-way ANOVA with Bonferroni's post hoc test, $p$ values of 2-HOBA vs vehicle and 2-HOBA vs 4-HOBA are ***<0.0001. Source data are provided as a source data file.

decreased in 2-HOBA treated $Ldlr^{-/-}$ mice (Fig. 2). That the effects of 2-HOBA are mediated by their action as dicarbonyl scavengers is further supported by the result that 4-HOBA, a geometric isomer of 2-HOBA, which is not an effective scavenger in vitro, is not atheroprotective and by the finding that MDA- and IsoLG-2-HOBA were abundantly formed vs -4-HOBA adducts (Supplementary Figs. 6, 7 and Supplementary Table 1) in hypercholesterolemic $Ldlr^{-/-}$ mice. In addition, the levels of urine F2-IsoP were not significantly different between 2-HOBA and 4-HOBA treated $Ldlr^{-/-}$ mice suggesting that the atheroprotective effects are not via inhibition of lipid peroxidation or chelating metal ions (Supplementary Fig. 8). A limitation to our

study is that the pharmacokinetics of 4-HOBA differ somewhat from that of 2-HOBA. While initial plasma concentrations after oral or intraperitoneal distribution do not significantly differ, elimination of 4-HOBA from the plasma compartment occurs more rapidly than for 2-HOBA. These differences in clearance raise the possibility that our finding that 4-HOBA dicarbonyl adducts were very low to undetectable in vivo could be due in part to the lower concentrations of 4-HOBA in tissues. However, it is important to note that the liver, spleen, and kidney contained similar levels of 2-HOBA vs 4-HOBA 30 min after intraperitoneal dosing (Supplementary Fig. 5). Furthermore, the aorta and heart levels of 2-HOBA and 4-HOBA were similar 30 min after oral

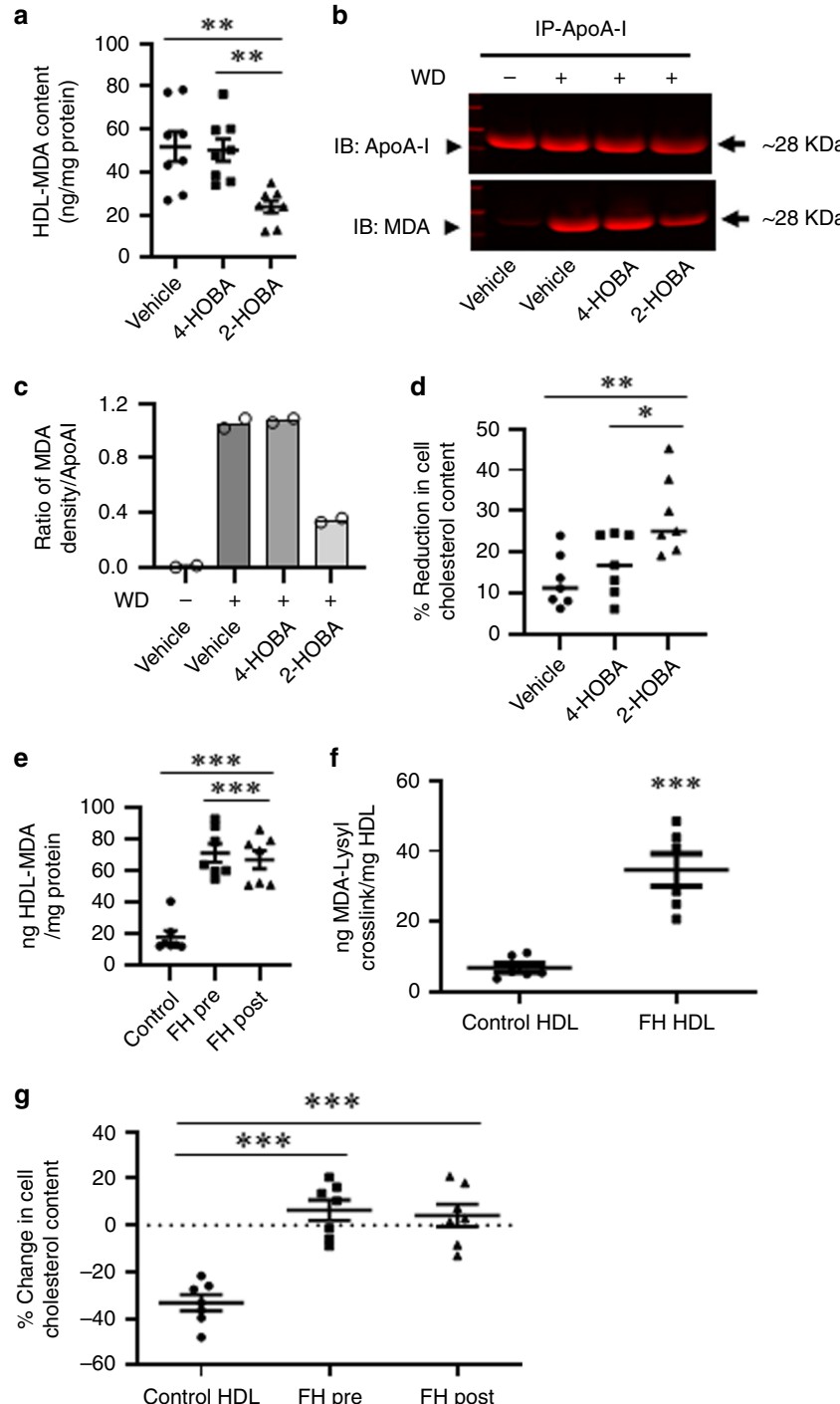

gavage of $Ldlr^{-/-}$ mice (Supplementary Fig. 4), suggesting equal access to scavenge reactive dicarbonyls in the developing atherosclerotic lesion. Our previous in vitro studies demonstrated poor reactivity of 4-HOBA vs 2-HOBA with reactive dicarbonyls[21]. Consistent with the lack of reactivity of 4-HOBA with reactive dicarbonyls in biological systems, when macrophages were treated in vitro with ox-LDL in the presence of 2-HOBA or 4-HOBA, 2-HOBA-MDA adducts were readily detected, whereas 4-HOBA-MDA adducts were undetectable (Supplementary Fig. 11). The concept that 2-HOBA vs 4-HOBA is an efficient in vivo scavenger of reactive dicarbonyls is substantiated by the finding that 19-fold more MDA-2-HOBA adducts accumulated in the urine during 16 h post oral gavage of $Ldlr^{-/-}$ mice (Supplementary Fig. 7A).

The increased levels of MDA-2-HOBA vs MDA-4-HOBA adducts in liver, spleen, and kidney 16 h after oral gavage of $Ldlr^{-/-}$ mice also strongly support that 2-HOBA is an effective in vivo dicarbonyl scavenger but 4-HOBA is not. Taken together, our studies showing that atherosclerosis can be prevented by utilizing 2-HOBA to remove dicarbonyls strengthens the hypothesis that reactive dicarbonyls contribute to the pathogenesis of atherogenesis and raises the therapeutic potential of dicarbonyl scavenging in atherosclerotic CVD. In this regard, we found that $Ldlr^{-/-}$ mice treated with 1 g of 2-HOBA/L of water had plasma levels of 2-HOBA that were similar to humans receiving oral doses of 2-HOBA in our recent safety trial in humans[24].

**Fig. 7 Effects of 2-HOBA on MDA-HDL adducts and HDL function. a** The levels of MDA adducts were measured by ELISA in HDL isolated from $Ldlr^{-/-}$ mice treated as described in Fig. 1. $n = 8$ biologically independent mice per group. Data are presented as mean ± SEM. One-way ANOVA with Bonferroni's post hoc test, $p$ values of 2-HOBA vs vehicle and 2-HOBA vs 4-HOBA are **0.0031 and **0.0053. Western blots (**b**) and quantitation (**c**) of apoAI and MDA-apoAI in HDL isolated from plasma by immunoprecipitation using primary anti-apoAI antibody. $Ldlr^{-/-}$ mice were treated as described in Fig. 1 and apoAI and MDA-apoAI from $Ldlr^{-/-}$ mice consuming a chow diet are included for comparison. Plasma was pooled from three mice per group from 2 experiments. **c** Quantitation of the mean density ratio (arbitrary units) of MDA-apoAI to ApoAI was done using ImageJ software $n = 2$ biologically independent experiments per group. Data are presented as the mean. **d** The HDL was isolated from the plasma of $Ldlr^{-/-}$ mice consuming a western diet for 16 weeks and treated with 2-HOBA or 4-HOBA or vehicle. Cholesterol-enriched macrophages were incubated for 24 h with HDL (25 μg protein/ml), and the % reduction in cellular cholesterol content measured. $n = 7$ biologically independent mice per group. Data are presented as mean ± SEM. One-way ANOVA with Bonferroni's post hoc test, $p$ values of 2-HOBA vs vehicle and 2-HOBA vs 4-HOBA are **0.0046 and *0.0375. **e** The MDA adducts were measured by ELISA in HDL isolated from control or FH subjects pre- and post-LDL apheresis. $n = 7$ biologically independent humans per group. Data are presented as mean ± SEM. One-way ANOVA with Bonferroni's post hoc test, $p$ values of control vs FH pre LDL apheresis and control vs FH post-LDL apheresis are ***<0.0001. **f** The MDA-Lysyl crosslink content in HDL from control or FH subjects. $n = 6$ biologically independent humans per group. Data are presented as mean ± SEM. Two-sided unpaired $t$ test, $p$ value of control vs FH-HDL is ***0.0002. **g** The capacity of HDL from control or FH subjects pre- and post-LDL apheresis to reduce the cholesterol content of apoE$^{-/-}$ macrophages. $n = 7$ independent humans per group. Data are presented as mean ± SEM. One-way ANOVA with Bonferroni's post hoc test, $p$ values of control vs pre LDL apheresis and control vs post-LDL apheresis are ***<0.0001. Source data are provided as a source data file.

HDL mediates a number of atheroprotective functions and evidence has mounted that markers of HDL dysfunction, such as impaired cholesterol efflux capacity, may be a better indicator of CAD risk than HDL-C levels[1,7,44–46]. Patients with FH have previously been shown to have impaired HDL cholesterol efflux capacity, indicative of dysfunctional HDL[47,48]. Our studies show that consumption of a western diet by $Ldlr^{-/-}$ mice results in enhanced MDA-apoAI adduct formation (Fig. 7), and that 2-HOBA treatment dramatically reduces modification of both apoAI and HDL with MDA. Similarly, FH patients had increased plasma levels of MDA-HDL adducts. In addition, in vitro modification of HDL with MDA resulted in decreased net cholesterol efflux capacity, similar to what we showed previously with IsoLG[43], and these effects were observed with HDL containing MDA adducts in the same range as FH subjects and hypercholesterolemic mice (Fig. 7 and Supplementary Fig. 15). Our results do not agree with other studies showing that MDA modification of HDL does not significantly impact cholesterol efflux from cholesterol-enriched P388D$_1$ macrophages, which may be due to differences in modification conditions or cell type[34]. Our findings are consistent with studies by Shao et al. demonstrating that modification of lipid-free apoAI with MDA blocks ABCA1 mediated cholesterol efflux[33]. In addition, studies have shown that long term cigarette smoking causes increased MDA-HDL adduct formation, and smoking cessation leads to improved HDL function with increased cholesterol efflux capacity[49]. In line with these results, we found that HDL isolated from 2-HOBA vs vehicle and 4-HOBA treated mice has enhanced capacity to reduce cholesterol stores in macrophage foam cells (Fig. 7). In addition, HDL from human subjects with FH had markedly increased MDA adducts and severely impaired ability to reduce macrophage cholesterol stores pre- and post-LDL apheresis (Fig. 7). Thus, one of the atheroprotective mechanisms of 2-HOBA is likely through preventing formation of dicarbonyl adducts of HDL proteins, thereby preserving HDL net cholesterol efflux function. In addition to decreasing HDL oxidative modification, our studies show that 2-HOBA treatment decreases the in vivo MDA modification of plasma LDL. Studies have shown that MDA modification of LDL promotes uptake via scavenger receptors resulting in foam cell formation and an inflammatory response[50,51]. The finding that incubation of macrophages with LDL from both 2-HOBA and 4-HOBA treated mice resulted in a similar cholesterol content is consistent with LDL, which is modified with sufficient amounts of MDA, being rapidly removed via scavenger receptors. However, studies have shown that neutralization of MDA-apoB adducts with antibodies greatly

enhances atherosclerosis regression in human apoB100 transgenic $Ldlr^{-/-}$ mice[52,53] making it likely that the decreased atherosclerosis with 2-HOBA treatment is also due in part to decreased dicarbonyl modification of apoB within the atherosclerotic lesion. Although, $Ldlr^{-/-}$ mice represent a relevant murine model of FH, there are limitations to the extent that our results from treatment of $Ldlr^{-/-}$ mice with 2-HOBA can be extrapolated to the anticipated therapeutic results in humans with FH or coronary artery disease. However, given that the initial phase I studies have demonstrated the safety of 2-HOBA in humans[24,54], we believe that our results support the importance of future translational studies to evaluate the impact of 2-HOBA on lipoprotein modification and function in humans with FH.

Evidence has mounted that increased oxidative stress in arterial intima cells is pivotal in inducing ER stress, inflammation, and cell death in atherogenesis[55]. In particular, efficient efferocytosis and limited cell death are critical to preventing the necrosis and excessive inflammation characteristic of the vulnerable plaque[1,56]. Our results demonstrate that treatment with 2-HOBA promotes characteristics of more stable atherosclerotic plaques in $Ldlr^{-/-}$ mice (Fig. 3). Consistent with this possibility, our study shows that 2-HOBA treatment decreased the atherosclerotic lesion MDA and IsoLG adduct content (Fig. 2), supporting the ability of dicarbonyl scavenging in the arterial intima to limit oxidative stress-induced inflammation, cell death, and destabilization of the plaque. Our studies show that scavenging of dicarbonyls with 2-HOBA in vitro limits oxidative stress-induced apoptosis in both endothelial cells and macrophages (Fig. 6). The decreased cell death is likely due in part to the greatly diminished inflammatory response to oxidative stress from dicarbonyl scavenging with 2-HOBA, as evidenced by the dramatic reductions in serum inflammatory cytokines including IL-1β (Fig. 5). An important limitation of these in vitro studies is that a relatively high concentration of 2-HOBA (500 μM) was used for most of these experiments, in part because we used a concentration of $H_2O_2$ (250 μM) to maximize the inflammatory response and apoptosis induced in these studies. While similar 2-HOBA concentrations can be achieved immediately following gavage of the entire daily intake of 2-HOBA (5 mg, as in Supplementary Fig. 4), steady state concentrations are much lower (i.e. 3–5 μM 2-HOBA) when the ~5 mg dose is administered by drinking water throughout the day as was done in our primary in vivo studies[22]. However, it is important to note our finding that 5 μM 2-HOBA sufficed to significantly reduce macrophage inflammatory cytokine expression in macrophages when a lower concentration of $H_2O_2$ (100 μM) was applied (Supplementary Fig. 10). These results are

particularly relevant given the recent results of the CANTOS trial showing that reducing inflammation with canakinumab, an IL-1β neutralizing monoclonal antibody, can reduce cardiovascular event rates in humans with prior MI and elevated hsCRP[57]. Importantly, treatment with 2-HOBA did not impact levels of urinary prostaglandin metabolites of prostacyclin, thromboxane, PGE2, and PGD2, indicating that 2-HOBA does not result in significant inhibition of COX in mice in vivo (Supplementary Fig. 13). Furthermore, 2-HOBA treatment maintained efficient efferocytosis and reduced the number of dead cells in the atherosclerotic lesions (Fig. 4). As a result, dicarbonyl scavenging with 2-HOBA promoted features of stable plaques with decreased necrosis and enhanced collagen content and fibrous cap thickness (Fig. 3). Hence, the ability of 2-HOBA to limit death and inflammation in arterial cells in response to oxidative stress and to promote efficient efferocytosis in the artery wall provides an atheroprotective mechanism whereby dicarbonyl scavenging promotes features of plaque stabilization and reduces atherosclerotic lesion formation. Our results are substantiated by recent studies demonstrating that $Ldlr^{-/-}$ mice expressing the single-chain variable fragment of E06 antibody to oxidized phospholipid have decreased atherosclerosis with stable plaque features including decreased necrosis and systemic inflammation[58], effects that are likely due in part to neutralization of esterified reactive dicarbonyls. Given the findings that significant residual inflammatory risk for CAD clinical events in humans independent of cholesterol lowering[57,59], these studies highlight reactive dicarbonyls as a target to decrease this risk. The prevention of atherosclerotic lesion formation is clearly an important strategy for the prevention of cardiovascular events. A limitation of the current study is that we have not examined the impact of 2-HOBA as an intervention for established lesions. In this regard, our future studies will be directed at examining whether reactive dicarbonyl scavenging can remodel established atherosclerotic lesions in $Ldlr^{-/-}$ mice.

In conclusion, 2-HOBA treatment suppresses atherosclerosis development in hypercholesterolemic $Ldlr^{-/-}$ mice. The atheroprotective effects of 2-HOBA likely result from preventing dicarbonyl adduct formation with plasma apoproteins and intimal cellular components. Treatment with 2-HOBA decreased the formation of MDA-apoAI adducts thereby maintaining efficient HDL function. In addition, the prevention of MDA-apoB adducts decreases foam cell formation and inflammation. Finally, within the atherosclerotic lesion, dicarbonyl scavenging limited cell death, inflammation, and necrosis thereby effectively promoting characteristics of stable atherosclerotic plaques. As the atheroprotective effect of 2-HOBA treatment is independent of any action on serum cholesterol levels, 2-HOBA offers real therapeutic potential for decreasing the residual CAD risk that persists in patients treated with HMG-CoA reductase inhibitors.

## Methods

**Mice.** $Ldlr^{-/-}$ and WT on C57BL/6 background mice were obtained from the Jackson Laboratory. The animal studies complied with all relevant ethical regulations for vertebrate animal research. Animal protocols were approved by and performed according to the regulations of Vanderbilt University's Institutional Animal Care and Usage Committee. Mice were maintained on chow or a Western-type diet containing 21% milk fat and 0.15% cholesterol (Teklad). Eight-week-old female $Ldlr^{-/-}$ mice on a chow diet were pretreated with vehicle alone (Water) or containing either 1 g/L of 4-HOBA or 1 g/L of 2-HOBA. 4-HOBA (as hydrochloride salt) was kindly provided by Dr. Davies[21]. 2-HOBA (as the acetate salt, CAS 1206675-01-5) was manufactured by TSI Co., Ltd (Missoula, MT) and obtained from Metabolic Technologies, Inc., Ames, IA[24]. A commercial production lot was used (Lot 16120312), and the purity of the commercial lot was verified to be >99% via HPLC and NMR spectroscopy[24]. After 2 weeks, the mice continued to receive these treatments but were switched to a western diet for 16 weeks to induce hypercholesterolemia and atherosclerosis. Similarly, 12-week-old male $Ldlr^{-/-}$ mice were pretreated with vehicle alone (water) or containing 1 g/L of 2-HOBA for two weeks and were then switched to a western diet for 16 weeks to induce

hypercholesterolemia and atherosclerosis, while continuing the treatment with 2-HOBA or water alone[60–62]. Based on the average weight and daily consumption of water per mouse the estimated daily dosage with 1 g/L of 2-HOBA is 200 mg/kg. We did not observe differences in mouse mortality among the treatment groups. Eight-week-old male $Ldlr^{-/-}$ mice were fed a western diet for 16 weeks and were continuously treated with either 2-HOBA or 4-HOBA. Urine samples were collected using metabolic cages (2 mice in one cage) during 18 h after oral gavage with either 2-HOBA or 4-HOBA (5 mg each mouse).

**Cell culture.** Peritoneal macrophages were isolated from mice 72 h post injection of 3% thioglycollate and maintained in DMEM plus 10% fetal bovine serum (FBS, Gibco)[32]. Human aortic endothelial cells were obtained from Lonza and maintained in endothelial cell basal medium-2 plus 1% FBS and essential growth factors (Lonza).

**Plasma lipids and lipoprotein distribution analyses.** The mice were fasted for 6 h, and plasma total cholesterol and triglycerides were measured by enzymatic methods using the reagents from Cliniqa (San-Macros, CA). Fast performance liquid chromatography (FPLC) was performed on an HPLC system model 600 (Waters, Milford, MA) using a Superose 6 column (Pharmacia, Piscataway, NJ).

**HDL isolation and measurement of cholesterol efflux.** HDL was isolated from mouse plasma using HDL Purification Kit (Cell BioLabs, Inc.) following the manufacturer's protocol. Briefly, apoB containing lipoproteins and HDL were sequentially precipitated with dextran sulfate. The HDL was then resuspended and washed. After removing the dextran sulfate, the HDL was dialyzed against PBS. To measure the capacity of the HDL to reduce macrophage cholesterol, $Apoe^{-/-}$ macrophages were cholesterol enriched by incubation for 48 h in DMEM containing 100 μg protein/ml of acetylated LDL. The cells were then washed, and incubated for 24 h in DMEM alone or with 25 μg HDL protein/ml. Cellular cholesterol was measured before and after incubation with HDL using an enzymatic cholesterol assay as described[63].

**Human blood collection and measurement of lipoprotein MDA.** The study protocol and informed consent forms were approved by the Vanderbilt Human Research Protection Program Institutional Review Board (IRB), and the study complies with all relevant ethical regulations for research with human participants. All participants gave their written informed consent in accordance with the Declaration of Helsinki. The human blood samples from patients with severe FH, who were undergoing LDL apheresis, and healthy controls were obtained using an IRB approved protocol. HDL and LDL were prepared from serum by Lipoprotein Purification Kits (Cell BioLabs, Inc.). Sandwich ELISA was used to measure plasma MDA-LDL and MDA-HDL levels following the manufacturer's instructions (Cell BioLabs, Inc.). Briefly, isolated LDL or HDL samples and MDA-Lipoprotein standards were added onto anti-MDA coated plates, and, after blocking, the samples were incubated with biotinylated anti-apoB or anti-ApoAI primary antibody. The samples were then incubated for 1 h with streptavidin-enzyme conjugate and 15 min with substrate solution. After stopping the reaction, the O.D. was measured at 450 nm wavelength. MDA-ApoAI was detected in mouse plasma by immunoprecipitation of ApoAI and western blotting. Briefly, 50 μl of mouse plasma was prepared with 450 μL of IP Lysis Buffer (Pierce) plus 0.5% protease inhibitor mixture (Sigma), and immunoprecipitated with 10 μg of polyclonal antibody against mouse ApoAI (Novus). Then 25 μL of magnetic beads (Invitrogen) was added, and the mixture was incubated for 1 h at 4 °C with rotation. The magnetic beads were then collected, washed three times, and SDS-PAGE sample buffer with β-mercaptoethanol was added to the beads. After incubation at 70 °C for 5 min, a magnetic field was applied to the Magnetic Separation Rack (New England), and the supernatant was used for detecting mouse ApoAI or MDA. For Western blotting, 30–60 μg of proteins was resolved by NuPAGE Bis-Tris electrophoresis (Invitrogen), and transferred onto nitrocellulose membranes (Amersham Bioscience). Membranes were probed with primary rabbit antibodies specific for ApoAI (Novus NB600-609) or MDA-BSA (Abcam cat# ab6463), GAPDH (Novus, cat# NB300-211) and fluorescent tagged IRDye 680 (LI-COR) secondary antibody. Proteins were visualized and quantitated by Odyssey 3.0 Quantification software (LI-COR).

**Modification of HDL and LDL with MDA.** MDA was prepared immediately before use by rapid acid hydrolysis of maloncarbonyl bis-(dimethylacetal) as described[33]. Briefly, 20 μL of 1 M HCl was added to 200 μL of maloncarbonyl bis-(dimethylacetal), and the mixture was incubated for 45 min at room temperature. The MDA concentration was determined by absorbance at 245 nm, using the coefficient factor 13, 700 $M^{-1}$ $cm^{-1}$. HDL (10 mg of protein /mL) and increasing doses of MDA (0, 0.125, 0.25, 0.5, and 1 mM) were incubated at 37 °C for 24 h in 50 mM sodium phosphate buffer (pH7.4) containing DTPA 100 μM. Reactions were initiated by adding MDA and stopped by dialysis of samples against PBS at 4 °C. LDL (5 mg/ mL) was modified in vitro with MDA (10 mM) in the presence of vehicle alone or with 2-HOBA at 37 °C for 24 h in 50 mM sodium phosphate buffer (pH7.4) containing DTPA 100 μM. Reactions were initiated by adding MDA and stopped by dialysis of samples against PBS at 4 °C. The LDL samples were incubated for

24 h with macrophages and the cholesterol content of the cells was measured using an enzymatic cholesterol assay as described[63].

**Atherosclerosis and immunofluorescence analyses**. The extent of atherosclerosis was examined in Oil-Red-O-stained cross-sections of the proximal aorta and by en face analysis. Briefly, cryosections of 10-micron thickness were cut from the region of the proximal aorta starting from the end of the aortic sinus and for 300 μm distally, according to the method of Paigen et al.[64]. The Oil-Red-O staining of 15 serial sections from the root to ascending aortic region were used to quantify the Oil-Red-O-positive staining area per mouse. The mean from the 15 serial sections was applied for the aortic root atherosclerotic lesion size per mouse using the KS300 imaging system (Kontron Elektronik GmbH) as described[65–67]. All other stains were done using sections that were 40–60 μm distal of the aortic sinus. For each mouse, four sections were stained and quantitation was done on the entire cross section of all four sections. For immunofluorescence staining, 5 μm cross-sections of the proximal aorta were fixed in cold acetone (Sigma), blocked in Background Buster (Innovex), incubated with indicated primary antibodies (MDA and CD68) at 4 °C for overnight. After incubation with fluorescent-labeled secondary antibodies at 37 C for 1 h, the nucleus was counterstained with Hoechst. Images were captured with a fluorescence microscope (Olympus IX81) and SlideBook 6 (Intelligent-Image) software and quantitated using ImageJ software (NIH).

**Analysis of apoptosis and efferocytosis**. Cell apoptosis was induced in vitro as indicated and detected by fluorescent-labeled Annexin V staining and quantitated by either Flow Cytometry (BD 5 LSRII, gating strategy is shown Supplementary Fig. 18) or counting Annexin V positive cells in images captured under a fluorescent microscope. The apoptotic cells in atherosclerotic lesions were measured by TUNEL staining of cross-sections of atherosclerotic proximal aortas and macrophages were detected by primary antibody FITC-CD68 (Abcam, Cat# 134351), the nucleus was counterstained with Hoechst. Images were captured with a fluorescence microscope (Olympus IX81) and SlideBook 6 (Intelligent-Image) software and quantitated using ImageJ software (NIH)[32]. The TUNEL-positive cells not associated with live macrophages were considered free apoptotic cells and macrophage-associated apoptotic cells were considered phagocytosed as a measure of lesion efferocytosis[32].

**Masson's trichrome staining**. Masson's trichrome staining was applied for measurement of atherosclerotic lesion collagen content, fibrous cap thickness and necrotic core size following the manufacture's instructions (Sigma)[32]. Briefly, 5 μm cross-sections of proximal atherosclerotic aorta root were fixed with Bouin's solution, stained with hematoxylin for nuclei (black) and biebrich scarlet and phosphotungstic/phosphomolybdic acid for cytoplasm (red), and aniline blue for collagen (blue). Images were captured and analyzed for collagen content, atherosclerotic cap thickness and necrotic core by ImageJ software[32]. The necrotic area is normalized to the total lesion area and is expressed as the % necrotic area.

**RNA isolation and real-time RT-PCR**. Total RNA was extracted and purified using Aurum Total RNA kit (Bio-Rad) according to the manufacturer's protocol. Complementary DNA was synthesized with iScript reverse transcriptase (Bio-Rad). Relative quantitation of the target mRNA was performed using specific primers, SYBR probe (Bio-Rad), and iTaqDNA polymerase (Bio-Rad) on IQ5 Thermocylcer (Bio-Rad) and normalized with 18S, as described earlier. The IL-1β, IL-6, TNF-α, and 18S primers are shown in Table 1.

**Analysis of urinary prostaglandin metabolites**. Concentrations of PGE-M, tetranor PGD-M, 11-dehydro-TxB[2] (TxB-M) and PGI-M in urine were measured in the Eicosanoid Core Laboratory at Vanderbilt University Medical Center. Urine (1 mL) was acidified to pH 3 with HCl. [$^2H_4$]-2,3-dinor-6-keto-PGF1a (internal standard for PGI-M quantification) and [$^2H_4$]-11-dehydro-TxB$_2$ were added, and the sample was treated with methyloxime HCl to convert analytes to the O-methyloxime derivative. The derivatized analytes were extracted using a C-18 Sep-Pak (Waters Corp. Milford, MA USA) and eluted with ethyl acetate. A [$^2H_6$]-O-methyloxime PGE-M deuterated internal standard was then added for PGE-M and PGD-M quantification. The sample was dried under a stream of dry nitrogen at 37 °C and then reconstituted in 75 μL mobile phase A for LC/MS analysis.

LC was performed on a 2.0 × 50 mm, 1.7 μm particle Acquity BEH C-18 column (Waters Corporation, Milford, MA, USA) using a Waters Acquity UPLC. Mobile phase A was 95:4.9:0.1 (v/v/v) 5 mM ammonium acetate:acetonitrile:acetic acid, and mobile phase B was 10.0:89.9:0.1 (v/v/v) 5 mM ammonium acetate:acetonitrile: acetic acid. Samples were separated by a gradient of 85–5% of mobile phase A over 14 min at a flow rate of 375 μl/min prior to delivery to a SCIEX 6500+ QTrap mass spectrometer.

Urinary creatinine levels are measured using a test kit from Enzo Life Sciences. The urinary metabolite levels in each sample are normalized using the urinary creatinine level of the sample and expressed in ng/mg creatinine.

**Measurement of 2-HOBA and 4-HOBA in plasma and tissue**. Measurement of 2-HOBA and 4-HOBA was performed by LC/MS after derivatization with phenylisothiocyanate (PITC), and using [$^2H_4$]-2-HOBA as an internal standard for 2-HOBA[22] (See Supplementary Fig. 16). For these assays, the Waters Xevo-TQ-Smicro triple quadrupole mass spectrometer operating in positive ion multiple reaction monitoring (MRM) mode monitored the following transitions: for PITC-2-HOBA or PITC-4-HOBA, m/z 259→107@20 eV (quantifier transition) and m/z 259→153@20 eV (qualifier transition); for PITC-[$^2H_4$]2-HOBA m/z 263→107@20 eV (quantifier transition), m/z 263→111@20 eV (qualifier transition). Abundance for PITC-2-HOBA was calculated based on the ratio of peak area vs that of PITC-[$^2H_4$]2-HOBA. One limitation of using [$^2H_4$]2-HOBA as an internal standard to measure 4-HOBA is that it required the use of an external calibration curve to calculate a correction factor. This method is not as accurate, but deuterated 4-HOBA was not available. Because the transition reactions for PITC-4-HOBA are less efficient than for PITC-2-HOBA, the ratio of peak areas for PITC-4-HOBA/PITC-[$^2H_4$]2-HOBA was multiplied by the correction factors 3.9 and 5.7 when using the m/z 107 and m/z 153 transition, respectively (see Supplementary Fig. 17).

**Measurement of IsoLG-Lys in aorta**. Isolation and LC/MS measurement of isolevuglandin-lysyl-lactam (IsoLG-Lys) adducts from aorta of 2-HOBA and 4-HOBA treated $Ldlr^{-/-}$ mice were performed using a Waters Xevo-TQ-Smicro triple quadrupole mass spectrometer[68].

**Detection of IsoLG adducts of 2-HOBA**. To generate an internal standard for quantitation, ten molar equivalents of the heavy isotope labeled 2-HOBA, [$^2H_4$]2-HOBA, were reacted with synthetic IsoLG[68] overnight in 1 mM triethylammonium acetate buffer to form IsoLG-2-HOBA adducts, and the adducts separated from unreacted [$^2H_4$]2-HOBA and IsoLG by solid phase extraction (Oasis HLB). The isolated reaction products of IsoLG-2-HOBA were scanned by mass spectrometer (Waters Xevo-TQ-Smicro triple quadrupole MS) operating in limited mass scanning mode to identify major products. In addition, precursor scanning with the product ion set at m/z 111.1 was used to confirm that the detected products were [$^2H_4$]2-HOBA adducts. Both methods showed that the primary adduct present in the purified IsoLG-[$^2H_4$]2-HOBA internal standard mixture was the IsoLG-[$^2H_4$]2-HOBA hydroxylactam adduct, although other adducts including pyrrole, lactam, and the anhydro- species of each of these adducts were also present. Similar species were seen when IsoLG was reacted with non-labeled 2-HOBA and precursor scanning using product ion m/z 107.1. To identify potential 2-HOBA adducts in tissue of treated animals, we first generated a list of 18 probable IsoLG-HOBA species [pyrrole, lactam, hydroxylactam based on the in vitro reactions of IsoLG and 2-HOBA and then the anhydro-, dinor-, dinor/anhydro-, tetranor-, and keto-(from oxidation of hydroxyl group) metabolites of each of these three adducts based on previous metabolism studies with prostaglandins and IsoP]. We then analyzed liver homogenate from a 2-HOBA treated mouse using LC/MS with the mass spectrometer operating in positive ion precursor scanning mode and the product ion set to m/z 107.1 and collision energy at 20 eV and looked for the presence of any of these precursor ions. Based on these data, we identified three potential metabolites: M1 precursor ion m/z 438.3, which mass is consistent with either the keto-pyrrole adduct or the anhydro-lactam adduct (both have identical mass). M2 m/z 440.3, which mass is consistent with the pyrrole adduct, and M3 m/z 454.3, which mass is consistent with the anhydro-hydroxylactam adduct or the keto-lactam adduct. When then sought to quantify the amount of the putative IsoLG-HOBA adducts in heart and liver samples as there was not sufficient aorta sample remaining from other analysis available to do this analysis.

For these experiments, liver or heart samples from $Ldlr^{-/-}$ mice treated with 2-HOBA or 4-HOBA were homogenized in 0.5 M Tris buffer solution pH 7.5

---

**Table 1 Real-time RT-PCR primer sequences used in this study.**

| Gene name | Forward primer 5′-3′ | Reverse primer 5′-3′ |
| --- | --- | --- |
| IL-1β | GGGCCTCAAAGGAAAGAATC | TACCAGTTGGGGAACTCTGC |
| IL-6 | GGGCCTCAAAGGAAAGAATC | TACCAGTTGGGGAACTCTGC |
| TNF-α | TATGGCTCAGGGTCCAACTC | CTCCCTTTGCAGAACTCAGG |
| 18S | AGTCCCTGCCCTTTGTACACA | CGATCCGAGGGCCTCACTA |

containing mixture of antioxidants (pyridoxamine, indomethacin, BHT, and TCEP). Total amount of protein in homogenate was determined for normalization. In total, 1 pmol IsoLG-[$^2$H$_4$]2-HOBA was then added to each homogenate sample as internal standard, the HOBA adducts extracted with ethyl acetate, dried, dissolved in solvent 1 (water with 0.1% acetic acid) and analyzed by LC/MS using Waters Xevo-TQ-Smicro triple quadrupole mass spectrometer operating in positive ion MRM mode, monitoring the following transitions: $m/z$ 438.3→107.1@20 eV for M1; $m/z$ 440→107.1@20 eV for M2; $m/z$ 454→107.1@20 eV for M3; and $m/z$ 476.3→111.1@20 eV for IsoLG-[$^2$H$_4$]2-HOBA hydroxylactam. Desolvation temperature: 500 °C; source temperature: 150 °C; capillary voltage: 5 kV, cone voltage: 5 V; cone gas flow 1 l/h; desolvation gas flow 1000 L/h. HPLC condition were as follows: Solvent 1: water with 0.1% acetic acid; Solvent 2, methanol with 0.1% acetic acid; column: Phenomenex Kinetex C8 50 × 2.1 mm 2.6 u 100 A, flow rate: 0.4 mL/min; gradient: starting condition 10% B with gradient ramp to 100% B over 3.5 min, hold for 0.5 min, and return to starting conditions over 0.5 min. Abundance for each metabolite was calculated based on the ratio of peak height vs that of internal standard.

**Analysis of dilysyl-MDA crosslinks by LC/ESI/MS/MS**. Samples (around 1 mg of protein) were digested with proteases for lysyl-lactam adducts[69]. Five nanograms of $^{13}$C$_6$-dilysyl-MDA crosslink standard were added to each cell sample and dilysyl-MDA crosslinks were purified[70]. The dilysyl-MDA crosslink was quantified by isotopic dilution by LC-ESI/MS/MS[70].

**LC/MS/MS quantification of scavenger-MDA adducts**. The scavenger-MDA adducts were extracted: (1) from homogenate of tissue (equivalent of 30 mg) or (2) from cells (1 ml), three times with 500 μl of ethyl acetate. The extract was dried down, resuspended in 100 μl of ACN-water (1:1, v/v with 0.1% formic acid), vortexed, and filtered through a 0.22 μm spin × column. The reactions were analyzed by LC-ESI/MS/MS using the column a Phenomenex Kinetex column at a flow rate of 0.1 ml/min. The gradient consisted of Solvent A, water with 0.2% formic acid and solvent B, acetonitrile with 0.2% formic acid. The gradient was as follows: 0–2 min 99.9% A, 2–9 min 99.9–0.1% A, and 9–12 min 99.9% B. The mass spectrometer was operated in the positive ion mode, and the spray voltage was maintained at 5000 V. Nitrogen was used for the sheath gas and auxiliary gas at pressures of 30 and 5 arbitrary units, respectively. The optimized skimmer offset was set at 10, capillary temperature was 300 °C, and the tube lens voltage was specific for each compound. SRM of specific transition ions for the precursor ions at $m/z$ 178→107 (propenal-HOBA adduct).

**Statistics**. Continuous data are summarized as mean ± SEM visualized by box plots and bar charts. Between-group differences were assessed with Student's $t$ test (2 groups) and one-way ANOVA (>2 groups, Bonferroni's correction for multiple comparisons). Their nonparametric counterparts, Mann–Whitney test (2 groups) and nonparametric Kruskal–Wallis test (more than 2 groups, Bunn's correction for multiple comparison) were used when assumptions for parametric methods were not met. The Shapiro–Wilk test was used to evaluate normality assumptions. All tests were considered statistically significance at two-sided significance level of 0.05 after correction for multiple comparisons. All statistical analyses were performed in GraphPad PRISM versions 5 or 7.

**Reporting summary**. Further information on research design is available in the Nature Research Reporting Summary linked to this article.

## Data availability

The data that support the findings of this study are available from the corresponding author upon reasonable request. Source data are provided with this paper.

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

## Acknowledgements

The analyses of urinary prostaglandin metabolites were performed in the Vanderbilt University Eicosanoid Core Laboratory. 2-HOBA was obtained from Metabolic Technologies, Inc., Ames, IA. This work was supported by National Institutes of Health Grants: HL116263 and DK59637 (Lipid, Lipoprotein, and Atherosclerosis Core of the Vanderbilt Mouse Metabolic Phenotype Centers).

## Author contributions

H.T. designed and performed key experiments, acquired and analyzed data, wrote the paper; J.H. designed and performed parts of experiments, acquired, and analyzed data; P.G.Y. designed research, analyzed data, and wrote the paper; J.L.B., Y.Z., L.D., V.Y., and I.Z.I. conducted parts of the experiments; F.Y. assisted with analysis of data and biostatistics; O.B., V.A., J.A.O., L.J.R.II, and S.S.D. designed and analyzed data and modified the paper; M.F.L. designed research, analyzed data, obtained funding, and wrote the paper.

## Competing interests

M.F.L., S.S.D., V.A., O.B., J.A.O., and L.J.R.II are inventors on a patent application for the use of 2-HOBA and related dicarbonyl scavengers for the treatment of cardiovascular disease. M.F.L. has received reseach funding from Amgen, Regeneron, Ionis, Merck, REGENXBIO, Sanofi and Novartis and has served as a consultant for Esperion, Alexion Pharmaceuticals and REGENXBIO. All the other authors (H.T., J.H., P.G.Y., J.L.B., F.Y., V.Y., V.Z., L.D., and I.Z.I.) declare no competing interests.
