## [Peer Review File · Nature Communications]

Reviewers' Comments:

Reviewer #1:

Remarks to the Author:

The oxidation hypothesis of atherosclerosis has been around since the early 80s and posited that oxidation of LDL led to enhanced uptake by macrophages causing foam cell formation, generation of a variety of oxidized lipid moieties and/or oxidized lipid-protein adducts which mediated enhanced inflammatory gene expression, cell death and atherogenesis. This remains to this day a central tenant of everyone's scheme of how atherogenesis occurs. The widespread acceptance of this hypothesis led to an investigation of a number of clinical trials using antioxidants (mostly for other purposes) as to their impact on cardiovascular disease. For the most part these utilized vitamin E or beta carotene and were mostly negative—for reasons discussed by you in the Introduction. In addition, a more fundamental reason for the failure to more properly test the oxidation hypothesis in humans is that we do not know the central molecular moieties that mediate the relevant oxidative events, nor the pathways by which they generate atherogenesis.

In this important new paper, seasoned and reliable investigators at Vanderbilt with expertise in atherogenesis and oxidative chemistry combine their efforts to convincingly demonstrate that a proven scavenger of reactive dicarbonyls can inhibit atherogenesis in western-diet fed Ldlr mice. In essence, the paper may be considered in two parts: Part 1: The impact of 2-HOBA on atherogenesis. These data are convincing and represent the heart of the paper—the use of the 4-HOBA is an excellent control and the data support reactive dicarbonyls as key mediators of atherosclerosis—an important advancement in our understanding of the role of oxidation in atherogenesis. Part 2: The authors go on to define the pathways by which 2-HOBA inhibits atherogenesis and present different lines of evidence to support multiple mechanisms by which scavenging dicarbonyls mediate this protective effect. Some of the mechanistic data are supportive and seemingly well done, but others seem much more tenuous. Atherogenesis is exceedingly complex and I recognize how difficult it is to develop definitive data regarding mechanisms, and I hope the authors will take the following comments in the constructive manner they are offered.

General comments: Que et al (Nature 2018) recently utilized mice expressing a single chain version of the E06 antibody that targets the PC of OxPL to demonstrate that OxPL were a key mediator of atherogenesis, promoting OxLDL uptake in macrophages and cholesterol accumulation, and proinflammatory gene expression. They demonstrated that targeting this oxidized moiety not only reduced lesion size, but similar to your findings, found this led to decreased necrotic cores of lesions and more stable lesion composition. There are many findings that are similar to your paper and it might be helpful to comment on this in your manuscript as they both offer substantial support for specific moieties in the pathogenesis of atherosclerosis. In particular, it might well be that some of the degradation products that are released when the unsaturated FA of the PL is oxidized may be some of the same dicarbonyls you are scavenging, such as MDA and others.

Specific comments:

The data on inhibiting atherogenesis is impressive. You should report on indices of health: the "weights were the same" what does this mean? When measured? Was the rate of weight gain the same? What about LFTs, etc. Among mice started on the different trials, were there any deaths? Imbalances between groups?

At the dose used in this study, 1 g/L, do you know what was the actual administered (consumed) dose in terms of mg/Kg and do you have any data on plasma levels and how these compare to doses found in humans administered 2-HOBA recently reported by Pitchford et al? This would be useful to understand the potential application to humans to inhibit dicarbonyl activity for the purposes of atherogenesis.

You report both en face and aortic root data for females but not males: the overall protective

effect was less in males—(whom most investigators find get more lesions than females)-- and no data are reported for en face, which is usually the easier measurement. Did you measure en face lesion changes? Were these decreased?

You report less "MDA levels" in aorta of 2-HOBA treated mice—but this is done by extent of MDA antibody binding—does 2-HOBA bind to MDA and inhibit the binding of the anti-MDA antibody? I presume most MDA in lesions after fixation are MDA-protein adducts –and likely 2-HOBA does not react with these? You previously published that isoketal adducts in the artery were decreased after 2-HOBA Rx of hypertension as I recall. It would be neat if you could show the same thing here as you have previously done this with a monoclonal antibody (Davies et al 2004).

What happened to MDA levels (either TBARS or mass spec) in plasma for example, aside from changes in MDA-LDL. Would you not expect these to decrease substantially?

You provide a series of experiments in vitro in which macrophages and EC are exposed to either H₂O₂ or OxLDL in the presence or absence of 2-HOBA and decreased apoptosis occurs and decreased inflammatory gene expression. How do you propose that 2-HOBA is doing this? What is the hypothesis? 2-HOBA does not bind H₂O₂ and therefore where are the dicarbonyls coming from to mediate the 2-HOBA effects. Similarly, what about OxLDL? Does 2-HOBA neutralize some direct products of OxLDL or is this an indirect effect on products of apoptotic cells?

Re impact on inflammation, the data in vitro are interesting but it would strengthen your suggestion that this is a relevant effect in vivo if you had evidence for an in vivo anti-inflammatory effect—either in macrophages or systemically? Changes in plasma MDA, SAA levels, changes in gene expression of peritoneal macrophages or inflammatory gene changes in artery or arterial macrophages or changes in plasma cytokines/chemokines to match in vitro data. Interestingly, there were no changes in COX pathway observed.

The changes in "MDA-LDL" and "MDA-HDL" are concerning. You coat an antibody to MDA on the wells and then add plasma and then measure apoB or apoA1 bound. This is only valid if the amount of LDL and apoA1 in particular added in each well is equal –I see the presumed LDL levels were similar from the FPLC and so possibly this was true if you add equal amounts of plasma. This is particular a concern for HDL, as not clear if composition same and if you added according to apoA1 content (and not HDL protein content), --was apoA1 measured and used to adjust the amount of HDL added to well? Especially true in the human FH patients for MDA-HDL. Further, if one LDL particle had one MDA adduct in one condition and 10 MDA's/LDL in the other—you would not be able to distinguish these conditions! Flipping the assay would be much more convincing—e.g. plating an anti-mouse apoB or apoA1 and then adding the anti-MDA antibody. In the absence of any further data, these qualifications must be discussed. Also, the legend for Fig 6A states: MDA-LDL adducts were measured by ELISA in Ldlr^{-/-} mice—technically I don't think you are measuring "MDA-adducts" as stated for reasons noted above.

What antibodies to mouse apoB and to mouse apoA1 were used, and similarly for humans?

The experiment showing that inclusion of 2-HOBA during MDA modification of LDL and hence decreased ability of the modified LDL to increase chol accumulation in macs is trivial—eg you prevent MDA modification of LDL and all this proves is that 2-HOBE binds MDA! How this relates to any in vivo setting is the issue: does the degree of MDA modification of a given LDL in vivo decrease sufficiently to inhibit uptake by macrophages? As mentioned above, the measurement of "MDA-LDL" does not provide this information.

Similarly, I don't understand how you are measuring MDA-apoA1 adducts! This is being done from western blots? For this to be valid you would need to show that each lane has equal amounts of apoA1 and only then would you be able to say that the MDA/apoA1 was changed. Please explain.

Measurement of extent of necrosis in lesions. Smaller lesions in general have smaller areas of

necrosis. Since the Rx rather dramatically decreases overall lesion size, I think you need to show that for lesions of similar size, the extent of necrosis is still decreased. This might be accomplished by normalizing the necrotic area of each section to the overall lesion size or ideally, by comparing necrotic lesion area in lesion sections of equal size. (You may be doing this already but can't tell from Methods)

What are the sources of the MDA and CD36 antibodies?

The whole issue of MDA modification of HDL and impact in vivo on cholesterol efflux has been dealt with in prior publications. The studies in FH patients are of considerable interest to support the relevance of MDA modification of HDL. However, again, details of how this was done need to be expanded—were these measurements again made by the western blot assay? In these same plasma, were measurements of “MDA-LDL” done? One would expect there to be similar elevation of this value no?

Reviewer #2:

Remarks to the Author:

The causative role of lipid peroxidation in experimental atherogenesis remains controversial. The present study addresses this issue by investigating the effects of 2-hydroxybenzylamine (2-HOBA) as a dicarbonyl scavenger on lesion size and characteristics, as well as plasma lipoprotein modification and function in female Ldlr^{-/-} mice fed a Western diet in a prevention setting. Overall, this is an interesting and mostly well-conducted study with several strengths, including the use of male mice, 4-HOBA as an inefficient dicarbonyl scavenger control, and the assessment of urinary prostaglandins. The findings presented have the potential to change current thinking about the role of oxidative stress in atherogenesis from primary (lipoprotein) lipid peroxidation and reactive 1-electron oxidants to secondary lipid oxidation with formation of reactive two electron oxidants (such as dicarbonyls). Notwithstanding this, there are a number of aspects where the present study could be strengthened.

Major Points:

1. Previous work has shown that 2-HOBA is an effective scavenger of dicarbonyls, and that 2-HOBA is more effective than 4-HOBA. Therefore, inclusion of treatment with 4-HOBA as a control is a clear strength of the present study, particularly as aminophenols (like HOBA) can also scavenge radical oxidants including peroxy radicals engaged in primary lipid peroxidation (defined as formation of lipid peroxides). The authors state (Introduction, third paragraph) that 2-HOBA and 4-HOBA are bioavailable (13, 22), although this reviewer could not find relevant information in ref. 13, and ref. 22 deals with 2-HOBA but not 4-HOBA in C57B/6J mice. To validate the use of 4-HOBA as a negative control, the authors need to demonstrate that plasma and lesion concentrations of both 2-HOBA and 4-HOBA are comparable in the animal model used here.
2. Related to Point 1, can the authors justify the concentrations of HOBA used for in vitro experiments based on in vivo drug concentrations?
3. If 2-HOBA protects by scavenging dicarbonyls (rather than by inhibiting lipid peroxidation), one would expect plasma and lesion concentrations of non-enzymatic primary lipid peroxidation products and markers thereof (such as F2-isoprostanes) to be comparable in vehicle control and 2-HOBA and 4-HOBA treated mice. Given the expertise of the authors in mass spectrometry-based analysis of F2-isoprostanes, this information should be provided, not least because it is central to the overall thesis put forward in the present study.
4. The results shown in Figure 2 is the exclusive evidence for 2-HOBA decreasing dicarbonyls in lesions, which is arguably more important than circulating dicarbonyls. As immunofluorescence is at best semi-quantitative, the study would gain in strength if data from a mass spectrometry based analysis of lesion dicarbonyls were provided, particularly as the authors have developed relevant methods (e.g., Anal Biochem 2019;566:89-101).
5. In the present study, 2-HOBA was administered before (and throughout) Western diet. What is

the impact of 2-HOBA in an intervention setting, i.e., when given after lesions have already developed? If such data is not available, the authors should list the prevention strategy used in the present study as a limitation.

6. While 2-HOBA did not change plasma cholesterol and triacylglycerols, information on the plasma and lesion content of the polyunsaturated fatty acid substrates for dicarbonyls is arguably more relevant and should be provided.

7. Is it difficult to know how precisely lesion and associated histological parameters were determined in the proximal aorta. The reference cited leads the reader on a chase to find specific relevant information, and one ends up with the 'Paigen method' for atherosclerosis assessment (Atherosclerosis 1987;68:231). It is important to provide details including where within the proximal aorta section were taken for the different types of staining, how many sections were analysed per animals, and what precisely was compared between treatment groups. Representative images in Figures 2-4 should (also) show entire cross sections.

Minor Points:

8. Can the authors explain why different doses of HOBA were used in male (3 g/L) compared with female mice 1 g/L)? Was water consumption measured or controlled for?

9. It is implied that Ldlr^{-/-} mice fed a Western diet form unstable plaque, when there is no convincing evidence for this or the reproducible occurrence of plaque rupture and associated processes (e.g., intra-plaque hemorrhage, thrombosis) in this model. Relevant text needs to be worded more carefully, just as Masson's Trichrome stain is a commonly used staining procedure for collagen – it does not analyse "lesion stability" (Figure 3, legend).

10. It is stated (e.g., in the Abstract) that oxidative stress accelerates atherogenesis, when this remains a hypothesis rather than a fact. Relevant text should be worded more carefully and an overall more balanced view should be provided to the reader.

11. It is argued that scavenging reactive lipid dicarbonyls does not interrupt normal ROS signalling and function, without providing relevant support, and when the validity of this assumption can be questioned in light of dicarbonyls being efficient activators of Nrf2 and related cellular responses.

12. The Y-axis label in Figure 6A refers to "HDL-MDA" when the figure appears to show data for LDL.

13. Individual data (rather than histograms) should be shown throughout. Similarly, if at all possible, please present data in absolute rather than relative terms (e.g., Figure 7D, E, G).

14. Please clarify what the results in Supplemental Figure 1 show. Is this pooled plasma, or representative of a single plasma analysis for each group (out of N=8)?

15. Please clarify which test was used for statistical analysis for each Figure and panel.

Roland Stocker

Reviewer #3:

Remarks to the Author:

The paper by Tao et al describes the ability of 2-hydroxybenzylamine (2-HOBA) to reduce atherosclerosis in hypercholesterolemic Ldlr^{-/-} mice. This pharmacological effect is explained by the Authors considering the effect of 2-HOBA in detoxifying di-carbonyl compounds, like malondialdehyde, which are involved in the onset and progression of atherosclerosis.

The paper is clear and well written and the pharmacological study demonstrates well the anti-atherogenic effect of 2-HOBA. The statistical analyses and the description of the materials and methods adequate.

However, I have some concerns about the novelty and the mechanism of action reported for 2-HOBA.

Several papers have so far reported the anti-atherogenic effect of compounds acting as sequestering agents of reactive carbonyl species such as, among others, pyridoxamine PMID:

21161164, aminoguanidine PMID: 15220206 and carnosine PMID: 20518851 PMID: 24468155 and derivatives PMID: 25471794 PMID: 23559625. On the basis of these studies, it is now fairly accepted that RCS are involved in the atherogenic response and that compounds effective as sequestering agents of RCS are effective in preventing atherosclerosis. RCS belong to different chemical classes (α,β -unsaturated aldehydes, di-aldehydes, keto-aldehyde and so on) and it is unclear whether all or some of them are involved in the atherogenic process. This open issue has been tentatively addressed by this paper which reports that 2-HOBA, a selective scavenger for di-carbonyls, as demonstrated elsewhere, is effective, thus indicating that di-carbonyls are involved as pathogenetic factors.

However, I believe that there is not enough data in the present paper to demonstrate that 2-HOBA acts as sequestering agent of di-carbonyls. The proposed mechanism is based only by indirect evidences such as the reduction of MDA LDL-content by treating with 2-HOBA and not by 4-HOBA (an isomer devoid of RCS sequestering activity). Additional data are required to confirm such a mechanism. Did the Authors search for the 2-HOBA adducts with MDA in tissue and/or biological fluids?. The reaction mechanism explaining the sequestering effect of 2-HOBA with di-aldehydes has been demonstrated in in vitro and homogeneous conditions and the reaction products elucidated, but to my knowledge no evidences that this reaction takes place in vivo exist.

In my opinion the mechanism of RCS sequestering agent is quite interesting but I am not very convinced that this reaction takes part in vivo because of the competitive reaction between the RCS scavenger and the nucleophilic substrates (proteins) which, although characterized by a reduced reaction constants in respect to the sequestering agents, are present in much higher concentrations. Kinetic measurements of the formation of adducts between 2-HOBA and MDA should be performed in biological matrices. In my opinion the paper could be of interest for a Nature Journal if the RCS sequestering activity of 2-HOPBA were demonstrated by direct and unequivocal evidences. Another possible mechanism of 2-HOBA activity could be related to a metal ion chelating effect. Did the Authors consider this and compare the metal ion chelating activity of 2-HOBA with that of 4-HOBA (most probably ineffective as metal ion chelator because it is a para isomer)?

Another important information would be to study the effect of 2-HOBA and 4-HOBA in cells by a quantitative proteomic approach to assess whether they have an effect on cell signalling and if they differentiate.

Response to the Reviewers' Comments

Reviewer #1 (Remarks to the Author):

The oxidation hypothesis of atherosclerosis has been around since the early 80s and posited that oxidation of LDL led to enhanced uptake by macrophages causing foam cell formation, generation of a variety of oxidized lipid moieties and/or oxidized lipid-protein adducts which mediated enhanced inflammatory gene expression, cell death and atherogenesis. This remains to this day a central tenant of everyone's scheme of how atherogenesis occurs. The widespread acceptance of this hypothesis led to an investigation of a number of clinical trials using antioxidants (mostly for other purposes) as to their impact on cardiovascular disease. For the most part these utilized vitamin E or beta carotene and were mostly negative—for reasons discussed by you in the Introduction. In addition, a more fundamental reason for the failure to more properly test the oxidation hypothesis in humans is that we do not know the central molecular moieties that mediate the relevant oxidative events, nor the pathways by which they generate atherogenesis.

In this important new paper, seasoned and reliable investigators at Vanderbilt with expertise in atherogenesis and oxidative chemistry combine their efforts to convincingly demonstrate that a proven scavenger of reactive dicarbonyls can inhibit atherogenesis in western-diet fed Ldlr mice. In essence, the paper may be considered in two parts: Part 1: The impact of 2-HOBA on atherogenesis. These data are convincing and represent the heart of the paper—the use of the 4-HOBA is an excellent control and the data support reactive dicarbonyls as key mediators of atherosclerosis—an important advancement in our understanding of the role of oxidation in atherogenesis. Part 2: The authors go on to define the pathways by which 2-HOBA inhibits atherogenesis and present different lines of evidence to support multiple mechanisms by which scavenging dicarbonyls mediate this protective effect. Some of the mechanistic data are supportive and seemingly well done, but others seem much more tenuous. Atherogenesis is exceedingly complex and I recognize how difficult it is to develop definitive data regarding mechanisms, and I hope the authors will take the following comments in the constructive manner they are offered.

We appreciate that the reviewer thought that our manuscript is relevant in that it supports reactive dicarbonyls as key mediators of atherosclerosis, which is an important advancement in our understanding of the role of oxidation in atherogenesis. We have addressed your insightful comments in detail below, and the manuscript has been revised accordingly. We believe that the revised manuscript has been substantially improved by addressing these comments.

General Comments:

Que et al (Nature 2018) recently utilized mice expressing a single chain version of the E06 antibody that targets the PC of OxPL to demonstrate that OxPL were a key mediator of atherogenesis, promoting OxLDL uptake in macrophages and cholesterol accumulation, and proinflammatory gene expression. They demonstrated that targeting this oxidized moiety not only reduced lesion size, but similar to your findings, found this led to decreased necrotic cores of lesions and more stable lesion composition. There are many findings that are similar to your paper and it might be helpful to comment on this in your manuscript as they both offer substantial support for specific moieties in the pathogenesis of atherosclerosis. In particular, it might well be that some of the degradation products that are released when the unsaturated FA of the PL is oxidized may be some of the same dicarbonyls you are scavenging, such as MDA and others.

We agree that the manuscript by Que and colleagues is relevant to our findings and have included comments on their results in the Discussion (Page 10).

Specific Comments:

1. The data on inhibiting atherogenesis is impressive. You should report on indices of health: the “weights were the same” what does this mean? When measured? Was the rate of weight gain the same? What about LFTs, etc. Among mice started on the different trials, were there any deaths? Imbalances between groups?

None of the *Ldlr*^{-/-} mice treated with either 2-HOBA or 4-HOBA died during the 18-week period, and we state this in the Methods section. There were also no significant differences in weight, diet intake, and water consumption between the treatment groups, which were measured weekly during the 16 weeks of consuming a western diet (Supplemental Figures 1A-1C).

2. At the dose used in this study, 1 g/L, do you know what was the actual administered (consumed) dose in terms of mg/Kg and do you have any data on plasma levels and how these compare to doses found in humans administered 2-HOBA recently reported by Pitchford et al? This would be useful to understand the potential application to humans to inhibit dicarbonyl activity for the purposes of atherogenesis.

Based on the average weight and water intake per day (1g 2-HOBA/L), each mouse is estimated to consume 200mg/Kg per day, and we now state this in the Methods section. Since our recent report, we have found that humans safely tolerate 750mg doses every 8h for 2 weeks. Based on an average weight of 75Kg, the dose administered each day is 40mg/Kg, which is in the same range used for our mouse studies assuming a mouse to human albumin dose conversion of $200/12.3 = 16.26$. The plasma levels of 2-HOBA in *Ldlr*^{-/-} mice treated with 1g 2-HOBA/L of water were 469 ± 38 ng/mL, which is in the same range as humans receiving 2-HOBA (*BMC Pharmacol Toxicol.* 2019; 20: 1).

3. You report both en face and aortic root data for females but not males: the overall protective effect was less in males—(whom most investigators find get more lesions than females)-- and no data are reported for en face, which is usually the easier measurement. Did you measure en face lesion changes? Were these decreased?

The atherosclerosis was decreased by 45% in enface aortas in male *Ldlr*^{-/-} mice, and the data is included in Supplemental Figures 2C and 2D).

4. You report less “MDA levels” in aorta of 2-HOBA treated mice—but this is done by extent of MDA antibody binding—does 2-HOBA bind to MDA and inhibit the binding of the anti-MDA antibody? I presume most MDA in lesions after fixation are MDA-protein adducts –and likely 2-HOBA does not react with these? You previously published that isoketal adducts in the artery were decreased after 2-HOBA Rx of hypertension as I recall. It would be neat if you could show the same thing here as you have previously done this with a monoclonal antibody (Davies et al 2004).

The antibody is from Abcam (cat# ab6463), and the immunogen was MDA conjugated to bovine serum albumin. We also provide data in Supplemental Figure 3 that demonstrates that the antibody does not recognize free MDA, and that the antibody also does not bind MDA-2-HOBA adducts. In addition, 2-HOBA does not interfere with the antibody recognition of MDA-albumin adducts. Furthermore, we have included measurements of aortic MDA- and IsoLG-lysine adducts by LC/MS/MS (Figures 2C and 2D), which demonstrate that treatment of *Ldlr*^{-/-} mice with 2-HOBA decreases these reactive dicarbonyl-protein adducts in atherosclerotic lesions compared to the mice treated with 4-HOBA.

5. What happened to MDA levels (either TBARS or mass spec) in plasma for example, aside from changes in MDA-LDL. Would you not expect these to decrease substantially?

We include data that show that treatment of *Ldlr*^{-/-} mice with 2-HOBA versus 4-HOBA or water alone decreased the plasma levels of MDA as measured using the TBAR assay (Supplemental Figure 11A).

6. You provide a series of experiments in vitro in which macrophages and EC are exposed to either H₂O₂ or OxLDL in the presence or absence of 2-HOBA and decreased apoptosis occurs and decreased inflammatory gene expression. How do you propose that 2-HOBA is doing this? What is the hypothesis? 2-HOBA does not bind H₂O₂ and therefore where are the dicarbonyls coming from to mediate the 2-HOBA effects. Similarly,

what about OxLDL? Does 2-HOBA neutralize some direct products of OxLDL or is this an indirect effect on products of apoptotic cells?

We now state clearly in the Results section that we propose that 2-HOBA decreases cell death and inflammation by scavenging reactive dicarbonyls that are formed from lipid peroxidation, when the cells are incubated with H₂O₂ or oxidized LDL. Studies have demonstrated that when cells are incubated with H₂O₂ lipid peroxidation occurs that results in the formation of reactive dicarbonyls including MDA and Isolevuglandin (*Biochem Pharmacol.* 1999;57:273-9, *Metab Brain Dis.* 2019 Jun 13. doi: 10.1007/s11011-019-00440-1, *Free Radic Res.* 2015;49:990-1003, and *Biochemistry.* 2006;45:15756-67). In addition, studies have shown that oxidized LDL induces generation of reactive oxygen species in cells, in particular, H₂O₂, which is necessary for the inflammatory response to oxidized LDL (*Circ Res.* 2009;104:210-8, *Redox Biol.* 2018;15:1-11, *Inflamm Res.* 2014;63:33-43, *J Cell Biochem.* 2017;118:661-669, and *J Clin Invest.* 2010 Nov;120:3996-4006). We also include new data in support of this hypothesis showing that, when macrophages are incubated with oxidized LDL, the inflammatory response is decreased in 2-HOBA versus 4-HOBA treated cells and MDA-2-HOBA adducts are formed during the incubation with 2-HOBA (Supplemental Figure 8).

7. Re impact on inflammation, the data in vitro are interesting but it would strengthen your suggestion that this is a relevant effect in vivo if you had evidence for an in vivo anti-inflammatory effect—either in macrophages or systemically? Changes in plasma MDA, SAA levels, changes in gene expression of peritoneal macrophages or inflammatory gene changes in artery or arterial macrophages or changes in plasma cytokines/chemokines to match in vitro data. Interestingly, there were no changes in COX pathway observed.

This is an excellent point and we include new data showing that treatment of *Ldlr*^{-/-} mice with 2-HOBA significantly reduces the plasma levels of IL-1 β , IL-6, TNF- α , and serum amyloid A versus 4-HOBA (Figure 5).

8. The changes in “MDA-LDL” and “MDA-HDL” are concerning. You coat an antibody to MDA on the wells and then add plasma and then measure apoB or apoA1 bound. This is only valid if the amount of LDL and apoA1 in particular added in each well is equal—I see the presumed LDL levels were similar from the FPLC and so possibly this was true if you add equal amounts of plasma. This is particular a concern for HDL, as not clear if composition same and if you added according to apoA1 content (and not HDL protein content), --was apoA1 measured and used to adjust the amount of HDL added to well? Especially true in the human FH patients for MDA-HDL. Further, if one LDL particle had one MDA adduct in one condition and 10 MDA's/LDL in the other—you would not be able to distinguish these conditions! Flipping the assay would be much more convincing—e.g. plating an anti-mouse apoB or apoA1 and then adding the anti-MDA antibody. In the absence of any further data, these qualifications must be discussed. Also, the legend for Fig 6A states: MDA-LDL adducts were measured by ELISA in *Ldlr*^{-/-} mice—technically I don't think you are measuring “MDA-adducts” as stated for reasons noted above.

We realize that the terminology we used to describe the measurement of MDA-LDL and MDA-HDL adducts in the Methods, Results, and Figure Legends is confusing and as such have rewritten the description to accurately reflect what was measured. The MDA content of LDL and HDL was measured using the MDA-LDL and MDA-HDL ELISA kits from Cell Biolabs. We did not add plasma to the wells coated with anti-MDA antibody. Isolated LDL or HDL were added to the antibody coated wells at the same protein concentration for different samples and MDA-modified LDL or HDL were used as standards. Each kit provides the reagents necessary to precipitate the lipoproteins and the modified lipoprotein standards. With the MDA-HDL kit, the HDL is isolated from the LDL and lipoprotein deficient fraction using a dual precipitation procedure similar to the methods described by Burstein and colleagues (*J Lipid Res.* 1970;11:583-595). We find that this method, when applying extra wash steps to the precipitated HDL pellet, yields HDL with the most abundant protein being apoAI as determined

by SDS PAGE. The immunogen for the MDA coating antibody in both kits was an MDA modified peptide containing lysine, and the antibody does not recognize free MDA. Taken together, we believe that utilizing this methodology provides measurements of MDA-LDL and MDA-HDL adducts. Consistent with the ELISA kit measuring MDA-HDL adducts, LC/MS/MS measurement of the most stable MDA adduct, N^ε-1-amino-3-iminopropenal crosslink of 2 lysines (HDL dilysyl-MDA crosslink), demonstrates a similar difference in control versus FH HDL dilysyl-MDA crosslinks (Figure 7F) with absolute values in the same range as the kit (Figure 7E), but lower, which likely results from quantitation of only one type of MDA adduct in the HDL.

9. What antibodies to mouse apoB and to mouse apoA1 were used, and similarly for humans?

The antibodies to apoA1 and apoB that were used for western blotting and/or immunoprecipitation were Novus NB600-609 and Santa Cruz Biotechnology SC-11795, respectively. Both antibodies react with human and mouse apoproteins. The anti-apoB and anti-apoA1 antibodies that are used with the Cell Biolab MDA-LDL and MDA-HDL ELISA kits also recognize both human and mouse apoproteins.

10. The experiment showing that inclusion of 2-HOBA during MDA modification of LDL and hence decreased ability of the modified LDL to increase chol accumulation in macs is trivial—eg you prevent MDA modification of LDL and all this proves is that 2-HOBA binds MDA! How this relates to any in vivo setting is the issue: does the degree of MDA modification of a given LDL in vivo decrease sufficiently to inhibit uptake by macrophages? As mentioned above, the measurement of “MDA-LDL” does not provide this information.

We agree with the reviewer and have now included data comparing the ability of LDL from 2-HOBA versus 4-HOBA treated *Ldlr*^{-/-} mice to enrich macrophages with cholesterol (Supplemental Figure 11D). There was no difference in the cholesterol content of macrophages incubated with LDL from 2-HOBA versus 4-HOBA treated *Ldlr*^{-/-} mice. We found that, while 2-HOBA does reduce the MDA content of LDL (Supplemental Figure 11A), the MDA adduct content of LDL from 4-HOBA treated *Ldlr*^{-/-} mice is not sufficient to increase the cholesterol content of macrophages.

11. Similarly, I don't understand how you are measuring MDA-apoA1 adducts! This is being done from western blots? For this to be valid you would need to show that each lane has equal amounts of apoA1 and only then would you be able to say that the MDA/apoA1 was changed. Please explain.

We have rewritten the description in the Methods and Figure Legends to better explain measurement of the MDA-ApoA1 by western blotting. First, apoA1 was immunoprecipitated from the same volume of plasma from the different treatment groups, and then the entire apoA1 fraction was loaded onto the gels. The membranes were first probed using the anti-MDA-BSA antibody (Abcam), where a strong signal was detected at the same molecular weight as apoA1, and then the membranes were probed for apoA1 (Figure 7B). The density of the MDA and apoA1 (Figure 7C) bands was then quantitated using Odyssey 3.0 Quantification software, and the data are expressed as the MDA band density/ApoA1 band density ratio (Figure 7C). The reviewer has a point in that it would be ideal to have equal amounts of apoA1 loaded onto the gels but probing of both MDA and apoA1 provides a means to correct for differences in apoA1 levels. It should be noted that in our study the 2-HOBA treated group had similar levels of apoA1 but the lowest level of MDA compared to 4-HOBA or vehicle treated mice (Figure 7B). In addition, we believe that the western blotting of MDA in the isolated HDL complements, and is in agreement with, the MDA-HDL adducts among the groups that were measured by ELISA (Figure 7A).

12. Measurement of extent of necrosis in lesions. Smaller lesions in general have smaller areas of necrosis. Since the Rx rather dramatically decreases overall lesion size, I think you need to show that for lesions of similar size, the extent of necrosis is still decreased. This might be accomplished by normalizing the necrotic

area of each section to the overall lesion size or ideally, by comparing necrotic lesion area in lesion sections of equal size. (You may be doing this already but can't tell from Methods)

The data in Figure 3 is the necrotic area normalized to the total lesion area and is expressed as the % necrotic area. We have now made this clear in the Methods on Page 12.

13. What are the sources of the MDA and CD36 antibodies?

We did not use CD36 antibodies and the sources of MDA-protein antibodies were Abcam and Cell Biolab as described above.

14. The whole issue of MDA modification of HDL and impact in vivo on cholesterol efflux has been dealt with in prior publications. The studies in FH patients are of considerable interest to support the relevance of MDA modification of HDL. However, again, details of how this was done need to be expanded—were these measurements again made by the western blot assay? In these same plasma, were measurements of “MDA-LDL” done? One would expect there to be similar elevation of this value no?

We agree that the impact of MDA modification of lipid-free apoAI on cholesterol efflux was clearly established in the studies by Shao and colleagues (*J Biol Chem* 2010;285:18473-18484). However, the effects of MDA modification of HDL on cholesterol efflux are less clear with the impact being minimal and nonsignificant in reducing the ³H cholesterol efflux in cholesterol enriched macrophages (*Free Radic Biol Med.* 1997;23:541-7), whereas other studies showed (*Biochim Biophys Acta.* 1992;1125:230-5) that there was a marked inhibition in the ³H cholesterol efflux from cholesterol-enriched fibroblasts. The differences in results with MDA modification of HDL may be related to the dose of MDA used or the cell type. In addition, we have found that with IsoLG modification of HDL versus lipid-free apoAI that different apoAI lysines are preferentially modified raising the possibility that different effects on function may occur with reactive dicarbonyl modification of lipidated apoAI compared to lipid-free apoAI. Given the controversial results with MDA modification of HDL and that the majority of plasma apoAI is lipidated, we believe that it is important to examine the effects of MDA modification of HDL on cholesterol efflux as it relates to our in vivo data with mouse HDL. In addition, we agree with the reviewer that the studies in FH patients are of interest and support the importance of MDA modification of HDL (Figures 7E and 7F). In this regard, we now include measurements of the MDA-HDL adducts formed when the HDL is modified with increasing doses of MDA (Supplemental Figure 12A) in addition to the effects on cholesterol efflux (Supplemental Figure 12B). Importantly, the data show that FH-HDL (Figures 7E) and HDL from vehicle and 4-HOBA treated *Ldlr*^{-/-} mice (Figure 7A) contain MDA adducts at levels that are sufficient to inhibit the ability to reduce macrophage cholesterol content. The MDA adducts (Figures 7A and 7E, and Supplemental Figure 12A) in the isolated HDL were measured using the ELISA kit as described above.

We measured the MDA adducts in LDL isolated from control and FH subjects by ELISA and determined that the MDA-LDL adducts were not different between the two groups (Supplemental Figure 11C). This observation is consistent with other studies showing that FH versus control LDL MDA content is not different (*Atherosclerosis.* 1995;118:259-73 and *Atherosclerosis.* 2003;166:261-70). There was also no difference in the cholesterol content of macrophages incubated with FH versus control LDL, which is consistent with the plasma LDL MDA content being too low to promote uptake (Data not shown).

Reviewer #2 (Remarks to the Author):

The causative role of lipid peroxidation in experimental atherogenesis remains controversial. The present study addresses this issue by investigating the effects of 2-hydroxybenzylamine (2-HOBA) as a dicarbonyl scavenger on lesion size and characteristics, as well as plasma lipoprotein modification and function in female *Ldlr*^{-/-}

mice fed a Western diet in a prevention setting. Overall, this is an interesting and mostly well-conducted study with several strengths, including the use of male mice, 4-HOBA as an inefficient dicarbonyl scavenger control, and the assessment of urinary prostaglandins. The findings presented have the potential to change current thinking about the role of oxidative stress in atherogenesis from primary (lipoprotein) lipid peroxidation and reactive 1-electron oxidants to secondary lipid oxidation with formation of reactive two electron oxidants (such as dicarbonyls). Notwithstanding this, there are a number of aspects where the present study could be strengthened.

We appreciate that the reviewer found our study to be interesting and have responded in detail to your comments. Our revised manuscript has new data that we believe has significantly improved the quality of the manuscript.

MajorPoints:

1. Previous work has shown that 2-HOBA is an effective scavenger of dicarbonyls, and that 2-HOBA is more effective than 4-HOBA. Therefore, inclusion of treatment with 4-HOBA as a control is a clear strength of the present study, particularly as aminophenols (like HOBA) can also scavenge radical oxidants including peroxy radicals engaged in primary lipid peroxidation (defined as formation of lipid peroxides). The authors state (Introduction, third paragraph) that 2-HOBA and 4-HOBA are bioavailable (13, 22), although this reviewer could not find relevant information in ref. 13, and ref. 22 deals with 2-HOBA but not 4-HOBA in C57B/6J mice. To validate the use of 4-HOBA as a negative control, the authors need to demonstrate that plasma and lesion concentrations of both 2-HOBA and 4-HOBA are comparable in the animal model used here.

The reviewer has an excellent point and we now provide evidence that both 2-HOBA and 4-HOBA are bioavailable. We measured the plasma levels of 2-HOBA versus 4-HOBA in *Ldlr*^{-/-} mice 30min after oral gavage (5mg) and found that the levels of 2-HOBA are 3.6-fold higher compared to 4-HOBA (Supplemental Figure 4A). We found this was due to increased clearance as we saw similar results in plasma of C57BL6 mice 30min after intraperitoneal injection of a 2.5mg (Supplemental Figure 4B). We recognize that the difference in clearance rate is a limitation in some respects, as it raises the possibility that our finding that there were undetectable (for IsoLG) or barely detectable (for MDA) 4-HOBA adducts in vivo might be due in part to the somewhat lower steady state levels of 4-HOBA compared to 2-HOBA. However, it is important to note that when macrophages were treated in vitro with ox-LDL in the presence of equivalent concentrations of 2-HOBA or 4-HOBA, 2-HOBA-MDA adducts were readily detected, whereas 4-HOBA-MDA adducts were undetectable (Supplemental Figure 8), supporting the lack of reactivity of 4-HOBA with the reactive dicarbonyl MDA in a biological system. While it might be theoretically desirable to match the 4-HOBA and 2-HOBA tissues levels by increasing the concentration of 4-HOBA in drinking water compared to 2-HOBA, we believe this would not be possible because the mice reduce their intake of water containing 4-HOBA at higher doses. Furthermore, any differences in clearance between 4-HOBA and 2-HOBA do not diminish the importance of the protective effects of 2-HOBA on atherosclerosis. In addition, the 2-HOBA effects were not due to general inhibition of lipid peroxidation as there were no differences in urine F₂-isoprostane levels (Supplemental Figure 7). It should be noted that the plasma levels of 2-HOBA shown to be atheroprotective in the male *Ldlr*^{-/-} mice after 16 weeks of treatment with 1g of 2-HOBA/L of water (469 ± 38 ng/mL) are in the same range as the plasma 2-HOBA levels in humans in our recent safety trial (*BMC Pharmacol Toxicol* **20**, 1 (2019)). We include these points in the Discussion (Page 8, lines 31-42).

2. Related to Point 1, can the authors justify the concentrations of HOBA used for in vitro experiments based on in vivo drug concentrations?

The amount used for the in vitro studies is in the range of plasma 2-HOBA levels 30 min after an oral dose of 2-HOBA (Supplemental Figure 4)

3. If 2-HOBA protects by scavenging dicarbonyls (rather than by inhibiting lipid peroxidation), one would expect plasma and lesion concentrations of non-enzymatic primary lipid peroxidation products and markers thereof (such as F2-isoprostanes) to be comparable in vehicle control and 2-HOBA and 4-HOBA treated mice. Given the expertise of the authors in mass spectrometry-based analysis of F2-isoprostanes, this information should be provided, not least because it is central to the overall thesis put forward in the present study.

The reviewer has an excellent point and measurement of the urinary F2-isoprostanes demonstrates that 2-HOBA is not preventing lipid peroxidation as the levels were similar between 2-HOBA and 4-HOBA treated *Ldlr*^{-/-} mice (Supplemental Figure 7).

4. The results shown in Figure 2 is the exclusive evidence for 2-HOBA decreasing dicarbonyls in lesions, which is arguably more important than circulating dicarbonyls. As immunofluorescence is at best semi-quantitative, the study would gain in strength if data from a mass spectrometry based analysis of lesion dicarbonyls were provided, particularly as the authors have developed relevant methods (e.g., *Anal Biochem* 2019;566:89-101).

We agree and have included measurements of aortic MDA- and IsoLG-lysine adducts by LC/MS/MS (Figures 2C and 2D). The data show that both adducts are decreased in aortas of 2-HOBA versus 4-HOBA treated *Ldlr*^{-/-} mice.

5. In the present study, 2-HOBA was administered before (and throughout) Western diet. What is the impact of 2-HOBA in an intervention setting, i.e., when given after lesions have already developed? If such data is not available, the authors should list the prevention strategy used in the present study as a limitation.

The prevention of atherosclerotic lesion formation is clearly an important strategy and goal for the development and testing of drugs for the prevention and treatment of atherosclerotic cardiovascular disease in animal models and in humans. We agree with the reviewer that an intervention study would also be of interest but consider this appropriate for future studies. The following statement is included in the Discussion (Page 10; lines 10-14). The prevention of atherosclerotic lesion formation is clearly an important strategy for the prevention of cardiovascular events. A limitation of the current study, is that we have not examined the impact of 2-HOBA as an intervention for established lesions. In this regard, our future studies will be directed at examining whether reactive dicarbonyl scavenging can remodel established atherosclerotic lesions in *Ldlr*^{-/-} mice.

6. While 2-HOBA did not change plasma cholesterol and triacylglycerols, information on the plasma and lesion content of the polyunsaturated fatty acid substrates for dicarbonyls is arguably more relevant and should be provided.

As it is conceivable that reactive dicarbonyl scavenging could affect plasma cholesterol by modulating reverse cholesterol transport and several studies have shown that plasma cholesterol levels definitely impact atherosclerosis development in *Ldlr*^{-/-} mice, we believe it was very important to measure plasma cholesterol levels. While we agree it is possible that dicarbonyl scavenging could affect levels of polyunsaturated fatty acid, we consider this beyond the scope of the current investigation given that the limited amounts of mouse plasma and aortic tissue were needed to measure other parameters. Furthermore, studies have shown that administration of polyunsaturated fatty acids does not affect atherosclerosis development in *Ldlr*^{-/-} mice (*Proc Natl Acad Sci U S A.* 2001;98(23):13294-9).

7. Is it difficult to know how precisely lesion and associated histological parameters were determined in the proximal aorta. The reference cited leads the reader on a chase to find specific relevant information, and one ends up with the 'Paigen method' for atherosclerosis assessment (*Atherosclerosis* 1987;68:231). It is important to provide details including where within the proximal aorta section were taken for the different types of staining,

how many sections were analysed per animals, and what precisely was compared between treatment groups. Representative images in Figures 2-4 should (also) show entire cross sections.

We now detail the method of atherosclerosis assessment in the Methods (Page 12) as follows: The hearts with the proximal aortas were harvested and embedded in OCT media and sectioned with a cryostat. Cryosections of 10-micron thickness were cut from the region of the proximal aorta starting from the end of the aortic sinus and for 300 μm distally, according to the method of Paigen et al. The Oil red-O staining of 15 serial sections from the root to ascending aortic region were used to quantify the Oil red-O-positive staining area per mouse. The mean from the 15 serial sections was applied for the aortic root atherosclerotic lesion size per mouse using the KS300 imaging system (Kontron Elektronik GmbH). All other stains were done using sections that were 40 to 60 μm distal of the aortic sinus. For each mouse, 4 sections were stained and quantitation was done on the entire cross section of all 4 sections. Representative entire cross section images are shown for the atherosclerosis analyses. For all other stains, representative higher magnification images are shown so that the stain is more clearly demonstrated and these are the same images that were used for quantitation with multiple images taken to cover the entire cross section. Dr. Linton is the Director of the Atherosclerosis Core of the Mouse Metabolic Phenotyping Center at Vanderbilt, and we now include additional references describing our approach to analyzing atherosclerosis in mice (Linton MF *Science* 1995; 267:1034-1037; Makowski L *Nature Medicine* 2001; 7: 699-705; Babaev VR *Cell Metabolism* 2008; 8:492-501).

8. Can the authors explain why different doses of HOBA were used in male (3 g/L) compared with female mice (1 g/L)? Was water consumption measured or controlled for?

Three g/L of 2-HOBA was used to determine how well the mice would tolerate the higher dose and if there was a greater reduction in atherosclerosis compared to 1g/L, and we have found that there is no difference in the consumption of water by mice treated with 1 g/L versus 3 g/L of 2-HOBA. We have removed the 3g/L data and now show data using 1g/L (Supplemental Figure 2) so that the impact of similar doses on atherosclerosis are shown for males and females.

9. It is implied that *Ldlr*^{-/-} mice fed a Western diet form unstable plaque, when there is no convincing evidence for this or the reproducible occurrence of plaque rupture and associated processes (e.g., intra-plaque hemorrhage, thrombosis) in this model. Relevant text needs to be worded more carefully, just as Masson's Trichrome stain is a commonly used staining procedure for collagen – it does not analyse “lesion stability” (Figure 3, legend).

We agree with the reviewer that the lack of plaque rupture in mice is a limitation of this widely used model. However, it is widely accepted in the atherosclerosis literature to characterize several features of mouse atherosclerotic lesions that are characteristic of unstable plaques in humans, such as the extent of necrosis, presence of inflammatory cells, collagen content and fibrous cap thickness. These types of lesion characterization are described as highly desirable by the recent Scientific Statement of the AHA on Animal Atherosclerosis Studies (Daugherty, *ATVB* 2017;37:e131–e157). We have made an effort to word related text more carefully.

10. It is stated (e.g., in the Abstract) that oxidative stress accelerates atherogenesis, when this remains a hypothesis rather than a fact. Relevant text should be worded more carefully and an overall more balanced view should be provided to the reader.

We have reworded the relevant text more carefully in an effort to provide a more balanced view.

11. It is argued that scavenging reactive lipid dicarbonyls does not interrupt normal ROS signaling and function, without providing relevant support, and when the validity of this assumption can be questioned in light of dicarbonyls being efficient activators of Nrf2 and related cellular responses.

The reviewer has a point and we have removed these statements from the text.

12. The Y-axis label in Figure 6A refers to “HDL-MDA” when the figure appears to show data for LDL.

We have corrected the axis label to the figure which is now Supplemental Figure 11A.

13. Individual data (rather than histograms) should be shown throughout. Similarly, if at all possible, please present data in absolute rather than relative terms (e.g., Figure 7D, E, G).

All data is presented as individual data points now, and where possible in absolute terms.

14. Please clarify what the results in Supplemental Figure 1 show. Is this pooled plasma, or representative of a single plasma analysis for each group (out of N=8)?

This is clarified in the Legend as follows: Plasma from 4 mice per group were pooled for each FPLC run and the average of duplicate FPLC runs for each mouse group are shown.

15. Please clarify which test was used for statistical analysis for each Figure and panel.
Roland Stocker

This information has been added to the Figure Legends.

Reviewer #3 (Remarks to the Author):

The paper by Tao et al describes the ability of 2-hydroxybenzylamine (2-HOBA) to reduce atherosclerosis in hypercholesterolemic Ldlr^{-/-} mice. This pharmacological effect is explained by the Authors considering the effect of 2-HOBA in detoxifying di-carbonyl compounds, like malondialdehyde, which are involved in the onset and progression of atherosclerosis.

The paper is clear and well written and the pharmacological study demonstrates well the anti-atherogenic effect of 2-HOBA. The statistical analyses and the description of the materials and methods adequate. However, I have some concerns about the novelty and the mechanism of action reported for 2-HOBA.

We appreciate the reviewer's comments and have added data to address the reviewer's concerns. We believe the additional data increases the novelty of the manuscript.

Major Points:

1. Several papers have so far reported the anti-atherogenic effect of compounds acting as sequestering agents of reactive carbonyl species such as, among others, pyridoxamine PMID: 21161164, aminoguanidine PMID: 15220206 and carnosine PMID: 20518851 PMID: 24468155 and derivatives PMID: 25471794 PMID: 23559625. On the basis of these studies, it is now fairly accepted that RCS are involved in the atherogenic response and that compounds effective as sequestering agents of RCS are effective in preventing atherosclerosis. RCS belong to different chemical classes (α , β -unsaturated aldehydes, di-aldehydes, keto-aldehyde and so on) and it is unclear whether all or some of them are involved in the atherogenic process. This open issue has been tentatively addressed by this paper which reports that 2-HOBA, a selective scavenger for

di-carbonyls, as demonstrated elsewhere, is effective, thus indicating that di-carbonyls are involved as pathogenic factors.

This is a valuable suggestion by the reviewer. We have revised the text of the manuscript to include the suggested papers with other classes of scavengers and the major carbonyls targeted by these scavengers. The inserted text is as follows (Page 7; Lines 33-50):

“A number of compounds with the potential to scavenge carbonyls have been identified, with individual compounds preferentially reacting with different classes of carbonyls so that the effectiveness of a scavenging compound in mitigating disease can serve as an indicator that their target class of carbonyl contributes to the disease process (*Curr Pharmacol Rep* 2017;3:1-67). Previous studies found that scavengers of methylglyoxal and glyoxal, such as aminoguanidine and pyridoxamine, reduce atherosclerotic lesions in streptozotocin-treated *Apoe*^{-/-} mice (*Diabetes* 2004;53:1813-1823 and *Diabetologia* 2011;54:681-689). Similarly, scavengers of α,β -unsaturated carbonyls (e.g. HNE and acrolein) such as carnosine and its derivatives, also reduce atherosclerosis in *Apoe*^{-/-} mice or streptozotocin-treated *Apoe*^{-/-} mice (*Arteriosclerosis, Thrombosis, and Vascular Biology* 2013;33:1162-1170, *Atherosclerosis* 2014;232:403-409, and *Diabetologia* 2015;58:845-853). These previously tested scavenger compounds are poor in vivo scavengers of lipid dicarbonyls such as IsoLG and MDA(*Curr Pharmacol Rep* 2017;3:1-67). Therefore, we sought to examine the potential of 2-HOBA, an effective scavenger of IsoLG and MDA, to prevent the development of atherosclerosis in *Ldlr*^{-/-} mice.

2. However, I believe that there is not enough data in the present paper to demonstrate that 2-HOBA acts as sequestering agent of di-carbonyls. The proposed mechanism is based only by indirect evidences such as the reduction of MDA LDL-content by treating with 2-HOBA and not by 4-HOBA (an isomer devoid of RCS sequestering activity). Additional data are required to confirm such a mechanism. Did the Authors search for the 2-HOBA adducts with MDA in tissue and/or biological fluids? The reaction mechanism explaining the sequestering effect of 2-HOBA with di-aldehydes has been demonstrated in in vitro and homogeneous conditions and the reaction products elucidated, but to my knowledge no evidences that this reaction takes place in vivo exist.

This is an excellent point by the reviewer, and we include new data showing that this reaction takes place in vivo. We have measured by LC/MS/MS (Supplemental Figure 6) the MDA- 2-HOBA adducts in urine and tissues from 2-HOBA versus 4-HOBA treated *Ldlr*^{-/-} mice. We found that the urine levels of propenal-HOBA adducts (Supplemental Figures 6A-6B) were increased by 20-fold in 2-HOBA versus 4-HOBA treated mice. In addition, the liver, kidney, and spleen from 2-HOBA versus 4-HOBA treated mice contained 3-, 5-, and 11-fold more propenal-HOBA adducts (Supplemental Figures 6C-6E). We have also determined that the levels of IsoLG-2-HOBA-lactam (keto metabolite), IsoLG-2-HOBA-pyrrole, and IsoLG-2-HOBA-anhydro-lactam are present in the heart and liver of 2-HOBA treated *Ldlr*^{-/-} mice (Supplemental Figure 5 and Table), but IsoLG- 4-HOBA adducts were not detectable in 4-HOBA treated *Ldlr*^{-/-} mice.

3. In my opinion the mechanism of RCS sequestering agent is quite interesting but I am not very convinced that this reaction takes part in vivo because of the competitive reaction between the RCS scavenger and the nucleophilic substrates (proteins) which, although characterized by a reduced reaction constants in respect to the sequestering agents, are present in much higher concentrations. Kinetic measurements of the formation of adducts between 2-HOBA and MDA should be performed in biological matrices. In my opinion the paper could be of interest for a Nature Journal if the RCS sequestering activity of 2-HOBA were demonstrated by direct and unequivocal evidences.

Our new data show that the reaction does take place in vivo (Supplemental Figures 5 and 6 and Table), and we have included additional data showing that in macrophages incubated with oxidized LDL, which

induces lipid peroxidation and inflammation, MDA-2-HOBA adducts are formed (Supplemental Figure 8). These results are consistent with our previous results demonstrating the formation of MDA-2-HOBA adducts in activated platelets incubated with 2-HOBA (*J Lipid Res.* 2015;56:2196-205).

4. Another possible mechanism of 2-HOBA activity could be related to a metal ion chelating effect. Did the Authors consider this and compare the metal ion chelating activity of 2-HOBA with that of 4-HOBA (most probably ineffective as metal ion chelator because it is a para isomer)?

While we agree that the metal ion chelating activity is likely greater with 2-HOBA compared to 4-HOBA, this effect would not explain our in vivo results as we did not observe a difference in urinary F₂-isoprostane levels (Supplemental Figure 7) in 2-HOBA versus 4-HOBA treated *Ldlr*^{-/-} mice.

5. Another important information would be to study the effect of 2-HOBA and 4-HOBA in cells by a quantitative proteomic approach to assess whether they have an effect on cell signaling and if they differentiate.

While we agree that proteomics analysis of the effects of 2-HOBA versus 4-HOBA on cell signaling could be interesting, we believe that such a broad analysis is beyond the scope of the current manuscript. We include data showing that there is no difference in phosphorylated Akt levels in 2-HOBA and 4-HOBA treated cells incubated in the absence of oxidative stress and with and without insulin (Supplemental Figure 9). These data suggest that there are no direct effects of 2-HOBA on anti-inflammatory signaling. In addition, when the cells were treated with 2-HOBA versus 4-HOBA in the presence of oxidative stress (Figure 6) to induce inflammatory pathways (*Circ Res.* 2009;104:210-8, *Redox Biol.* 2018;15:1-11, *Inflamm Res.* 2014;63:33-43, *J Cell Biochem.* 2017;118:661-669, and *J Clin Invest.* 2010;120:3996-4006), there is formation of MDA-2-HOBA adducts (Supplemental Figure 8). This suggests that any effects on inflammatory signaling are indirect via the scavenging of bioactive dicarbonyls.

Reviewers' Comments:

Reviewer #1:

Remarks to the Author:

The many responses of the authors have greatly improved the manuscript, which I feel helps to demonstrate a role for dicarbonyls in atherogenesis. The replies for the most part help support the authors' viewpoints but a number of observations revealed by the review diminish in part the enthusiasm.

Chief among these are the revelation that the mean plasma levels of 2-HOBA are 3.6 fold higher in plasma after only 30 min of gavage—a major strength of paper was assumption that all (or most) of differences seen were due to 2-HOBA specific effects. What were 2-HOBA levels vs 4-HOBA after being on diet for some time, and in particular on WD—where absorption may have been more affected. You state the plasma difference at 30 min due to clearance as you saw same difference after ip injection but again this could be due to differences in peritoneal binding/absorption rather than plasma clearance, and even if this is more rapid clearance we need to know steady state levels of both compounds respectively under conditions noted especially of WD. You state plasma levels of 2-HOBA 469 ng/ml. What were 4-HOBA levels (or did I miss this)

The changes in plasma MDA levels are helpful

What does Fig Supp 11 panel C show? What are FH pre and FH post

I continue to not understand the validity of the MDA-LDL or MDA-HDL measurements by ELISA other than qualitative effects.

The presence of MDA-2HOBA adducts are an important quantitative addition.

How do the authors explain lack of difference of F2 isoprostanes in urine as “evidence of lack of effect on lipid peroxidation”, yet what do they ascribe lower levels of MDA in plasma?

Point 15 of Reviewer 2? Please provide answer to this?

Point 5 of Reviewer 3: The authors show that there is no effect of 2-HOBA on macrophage anti-inflammatory signaling. Did 2-HOBA alter levels of plasma cytokines in chow fed mice? Obviously with 2-HOBA in vivo there were decreased plasma cytokine levels in WD fed mice and you ascribe all of this to MDA-2-HOBA adducts and thus indirect effects of scavenging MDA and other dicarbonyls.

Reviewer #2:

Remarks to the Author:

The additional comments below relate to the original review.

Major Points:

1. The additional experiments carried out with bolus administration of HOBA show significantly lower bioavailability of 4-HOBA than 2-HOBA, indicating that under the experimental conditions used 4-HOBA is not an appropriate control because the two compounds are not present in vivo at the same concentration. Separate to this, the additional experiments (Supplemental Figure 8) do not establish the suitability of 4-HOBA as an ineffective dicarbonyl scavenger control. This is because analysis of “propenal-HOBA adduct” is based entirely on SRM of the transition m/z 178 \diamond 107. While this transition is reasonably assigned to the propenal adduct of 2-HOBA, none of the experiments reported establishes that it is also appropriate for the detection of the propenal adduct of 4-HOBA. Therefore, the absence of “propenal-HOBA adduct” after exposure of macrophages to oxLDL plus 4-HOBA does not establish a lack of reaction of 4-HOBA with the

reactive bicarbonyl MDA in a biological system, and hence 4-HOBA as an inefficient dicarbonyl scavenger, just as the lack of in vivo protection by 4-HOBA cannot be taken as evidence for 2-HOBA mediating protection via dicarbonyl scavenging. Also, the additional studies do not justify the concentrations of HOBA used for in vitro experiments because the amount of HOBA administered by bolus (5 mg) corresponds to the amount of HOBA consumed over 24 h, with 1 g HOBA/L drinking water and assuming a consumption of 5 mL drinking water per day for a 30 g mouse (Physiol Behav 2007;91:620). This discrepancy in administered dose is reflected in the plasma concentration of 2-HOBA determined 30 min (single time point) after bolus addition ($\sim 40,000 \pm 10,000$ ng/mL; Supplemental Figure 4A) being 2 orders of magnitude higher than the steady state concentration of 2-HOBA determined 16 weeks after treatment with 1 g 2-HOBA per litre drinking water, i.e., 469 ± 38 ng/mL. As a result, there remain fundamental problems with the proposed mode of anti-atherosclerotic activity of 2-HOBA.

2. The concentration of HOBA used for in vitro experiments is $\sim 1,000$ -fold higher than the steady state 2-HOBA concentration in plasma under conditions where anti-atherosclerotic activities are observed. Even compared with 2-HOBA plasma concentrations observed acutely after a bolus administration of a daily dose of 2-HOBA (that does not reflect the steady-state concentrations of the inhibitor), the in vitro concentrations used are 10-fold higher. Based on these findings, it is doubtful that the in vitro experiments meaningfully recapitulate the in vivo conditions.

3. The urinary concentrations of F2-IPs observed (Supplementary Figure 7) suggest that 2-HOBA or 4-HOBA have no effect on systemic oxidative stress. Unfortunately, the new data does not provide direct information on whether 2- and 4-HOBA comparably affect the PUFA-adjusted concentrations of F2-IPs within lesions, the site where 2-HOBA is proposed to act as dicarbonyl scavenger.

4. It is appreciated that the authors have performed additional LC-MS analyses of lesion material to provide quantitatively more convincing evidence that 2-HOBA indeed acts as a dicarbonyl scavenger in atherosclerotic lesion. Unfortunately, however, the new data provided (Figures 2C and 2D) lack data for vehicle, so that it is not possible from this data to conclude that 2-HOBA decreases MDA- and IsoLG-protein adducts. Comparing MDA- and IsoLG-protein adducts in 4-HOBA versus 2-HOBA does not overcome this deficiency. This is because such data is complicated to interpret because it is not clear whether the MS/MS transitions used, based on fragmentation of IsoLG- and MDA-2-HOBA, are identical and hence relevant for IsoLG- and MDA-4-HOBA (see also comment 1 above). The additional LC-MS/MS data provided in Supplemental Figure 5 add further concern. Specifically, it is unclear why the internal standard [$^2\text{H}_4$]IsoLG-2HOBA added to livers of 2-HOBA treated mice gives an m/z 476.3 \diamond 111.1 signal nearly 10-times higher than that seen with livers from 4-HOBA treated animals. In addition, the protein adducts reported in new Figure 2C and 2D should be standardised to lipid rather than protein, because IsoLG and MDA are derived from lipid oxidation and the lesion lipid content is different in 2-HOBA versus 4-HOBA-treated mice.

5. It is appreciated that carrying out intervention studies is beyond the scope of the present investigation, so that stating the limitation of the present study is appropriate. Notwithstanding this, however, the authors' view that "the prevention of formation of atherosclerotic lesion is an important strategy for the prevention of cardiovascular disease" can be challenged. I am not aware of any drug-related strategy currently in use that is commenced before atherosclerotic lesions are present, and such strategy seems unlikely to be relevant in the near future, not least because treatment would have to commence in childhood and continue throughout life. Is this really what the authors have in mind for 2-HOBA, or am I missing something?

6. The point raised here is two-fold: First, dicarbonyls are derived from PUFA (and perhaps more precisely bisallylic hydrogen-containing lipids) so that the dicarbonyl content is likely affected by the PUFA content. Second, if 2-HOBA decreases lesion size, it is expected to decrease the pool of PUFA from which dicarbonyls are formed, so that adduct formation needs to be lipid standardised (see comment 4).

7. OK

Minor Points:

8. OK

9. OK
10. OK
11. OK
12. OK.
13. OK
14. OK
15. Non-parametric or parametric tests were used for statistical analyses of the data. Please justify why a specific statistical method was used (performing normality and equal variance tests for continuous variables).

Reviewer #3:

Remarks to the Author:

In the revised version the authors added novel results which have satisfied my concerns. In particular, the identification of adducts between 2-HOBA and di-aldehydes strengthens the carbonyl sequestering mechanism of the therapeutic agent, further sustained by the F2-isoprostanes results.

In my opinion the paper is of great interest because it reveals novel and important aspects of carbonylation stress and in particular identify di-carbonyls as pathogenetic factors of atherosclerosis. Moreover the paper furnishes convincing evidences of a potential drug target and of a new class of pharmacological tools. The present paper will lead to a further great scientific interest to carbonyl stress and to sequestering agents able to prevent it

Response to the Reviewers' Comments:

We appreciate the Reviewers careful review of our revised manuscript and their insightful comments. We are pleased that Reviewer 1 said, "The many responses of the authors have greatly improved the manuscript, which I feel helps to demonstrate a role for dicarbonyls in atherogenesis." In addition, Reviewer 3 commented, "In the revised version the authors added novel results which have satisfied my concerns. In particular, the identification of adducts between 2-HOBA and di-aldehydes strengthens the carbonyl sequestering mechanism of the therapeutic agent, further sustained by the F2-isoprostanes results. Furthermore, we appreciate the constructive comments from Reviewer 2 regarding the apparent differences in tissue levels of 2-HOBA and 4-HOBA. We have responded in detail to each of the Reviewer's comments, and we believe that addressing these comments has significantly improved our manuscript. We hope that you will agree that we have adequately addressed all of the issues that were raised.

Reviewer #1: (Remarks to the Author):

1) The many responses of the authors have greatly improved the manuscript, which I feel helps to demonstrate a role for dicarbonyls in atherogenesis. The replies for the most part help support the authors' viewpoints but a number of observations revealed by the review diminish in part the enthusiasm.

Chief among these are the revelation that the mean plasma levels of 2-HOBA are 3.6-fold higher in plasma after only 30 min of gavage—a major strength of paper was assumption that all (or most) of differences seen were due to 2-HOBA specific effects. What were 2-HOBA levels vs 4-HOBA after being on diet for some time, and in particular on WD—where absorption may have been more affected. You state the plasma difference at 30 min due to clearance as you saw same difference after ip injection but again this could be due to differences in peritoneal binding/absorption rather than plasma clearance, and even if this is more rapid clearance we need to know steady state levels of both compounds respectively under conditions noted especially of WD. You state plasma levels of 2-HOBA 469 ng/ml. What were 4-HOBA levels (or did I miss this).

We appreciate Reviewer 1's favorable comments, including the statement that "the manuscript ... helps to demonstrate a role for dicarbonyls in atherogenesis." A detailed response to these questions and new studies to address these questions are given in response to Reviewer #2, below. In summary, the levels of 4-HOBA were not determined correctly in the previous submission. Our new analysis demonstrates that the concentration of 4-HOBA does not significantly differ from that of 2-HOBA 30 min after oral gavage of *Ldlr*^{-/-} mice that had been on the WD for 16 weeks (Suppl. Fig. 4A). Because of the long duration of the WD, we would anticipate that any change in liver or enzyme levels of metabolic enzymes for 2-HOBA or 4-HOBA would have manifested themselves in these data. Furthermore, levels of 2-HOBA and 4-HOBA in the aortas and hearts of *Ldlr*^{-/-} mice 30 min after oral gavage are not significantly different (Suppl. Fig. 4B,C). However, 4-HOBA does undergo somewhat faster clearance than 2-HOBA, which we already knew has a short half-life of 62 min in mice (Suppl. Fig. 5A). The 4-HOBA levels in plasma after 16 weeks of WD administration were 25 ± 3 ng/mL. Given the short half-life of these compounds, this plasma level should not be viewed as "steady state". Mice are nocturnal and activity and water consumption are highest at night not during the day when the plasma samples were collected. A number of factors may have contributed to the lower plasma levels of 4-HOBA, including the increased rate of clearance, variation in water consumption, and the time at which the plasma samples were drawn. Additional experiments using intraperitoneal (IP) injection in chow fed mice show that the liver, spleen, and kidney levels of 2-HOBA versus 4-HOBA are not significantly different 30 min after injection (Suppl. Fig. 5B-D). Thus, when equal amounts of 2-HOBA and 4-HOBA are administered at the same time by the oral or IP route, the levels in a variety of tissues are not different after 30 minutes, demonstrating that both 2-HOBA and 4-HOBA are bioavailable to the tissues.

2) The changes in plasma MDA levels are helpful. What does Fig Supp 11 (now Suppl. Fig. 14) panel C show? What are FH pre and FH post

The MDA adducts were measured in LDL isolated from FH subjects pre- and post-LDL apheresis, and we revised the Suppl. Figure 14 Legend to clarify.

3) I continue to not understand the validity of the MDA-LDL or MDA-HDL measurements by ELISA other than qualitative effects.

The ELISA assays are quantitatively measuring the MDA-LDL and MDA-HDL adducts as isolated LDL and HDL are added to the anti-MDA coated plates, and the assays are standardized to MDA modified-LDL and -HDL where the MDA content has been determined by TBARS.

4) The presence of MDA-2-HOBA adducts are an important quantitative addition.

We agree that MDA-2-HOBA adducts as well as the IsoLG-2-HOBA adducts are important additions, which support our hypothesis that the effects of 2-HOBA on atherosclerosis is via scavenging reactive dicarbonyls.

5) How do the authors explain lack of difference of F2 isoprostanes in urine as “evidence of lack of effect on lipid peroxidation”, yet what do they ascribe lower levels of MDA in plasma?

Lipid peroxidation generates many products including F2 isoprostanes, isolevuglandins, and MDA. If 2-HOBA was acting to lower lipid peroxidation generally, it would reduce the levels of all of these. If instead, as we hypothesize, 2-HOBA acts as a dicarbonyl scavenger, we would expect that the levels of dicarbonyls like IsoLG and MDA and their adducts would be reduced, while non-dicarbonyl products of lipid peroxidation like F2-isoprostanes would not change. Given that these are the results we see, this provides support for the hypothesis that 2-HOBA acts primarily via dicarbonyl scavenging.

6) Point 15 of Reviewer 2? Please provide answer to this?

Please see detailed response to Point 15 of Reviewer 2 below.

7) Point 5 of Reviewer 3: The authors show that there is no effect of 2-HOBA on macrophage anti-inflammatory signaling. Did 2-HOBA alter levels of plasma cytokines in chow fed mice? Obviously with 2-HOBA in vivo there were decreased plasma cytokine levels in WD fed mice and you ascribe all of this to MDA-2-HOBA adducts and thus indirect effects of scavenging MDA and other dicarbonyls.

Reviewer 3 wanted to know if the scavenger was having a direct effect on signaling. We examined the impact of 2-HOBA on phosphorylation of Akt in macrophages in the absence of oxidative stress using the experimental conditions described for Suppl. Figure 12. No effect was observed on pAkt suggesting that 2-HOBA does not directly affect this anti-inflammatory signaling. Consistent with oxidative stress increasing in mice fed a western diet and with the effects of 2-HOBA being indirect on anti- and pro-inflammatory pathways via scavenging of reactive carbonyls, there were no differences in plasma levels of TNF- α , IL-6, and IL-1 β in chow fed 2-HOBA versus 4-HOBA treated mice (Suppl. Figure 9).

Reviewer #2 (Remarks to the Author):

The additional comments below relate to the original review.

Major Points:

1. The additional experiments carried out with bolus administration of HOBA show significantly lower bioavailability of 4-HOBA than 2-HOBA, indicating that under the experimental conditions used 4-HOBA is not an appropriate control because the two compounds are not present in vivo at the same concentration.

We thank the reviewer for raising this concern, as the studies that we undertook to address this concern (which

will be outlined more completely below) identified an issue that we had not originally recognized when we were measuring 4-HOBA levels in plasma and tissue. In essence, the levels of 4-HOBA we initially calculated from our mass spectrometry data failed to take into account differences in the ionization/fragmentation efficiency of the PITC derivative of 4-HOBA (PITC-4-HOBA) compared to the PITC derivative of 2-HOBA (PITC-2-HOBA). Once we identified this issue and determined the appropriate correction factor to account for this difference in efficiency, the calculated levels of 4-HOBA and 2-HOBA in plasma and tissue after treatment are actually fairly similar. For cohesiveness, all of the studies we conducted to more fully characterize important aspects of our analysis of 2-HOBA and 4-HOBA levels and their ADME characteristics will be laid out as one integrated section here and then we will repeat key sections in response to individual concerns.

Measurement of 2-HOBA and 4-HOBA adducts using MRM transitions to the m/z 107 product ion.

Reviewer 2 asked for evidence that the m/z 107 product ion was an appropriate product ion for monitoring 4-HOBA adducts/derivatives, and not just 2-HOBA adducts/derivatives. We have previously reported some of the 2-HOBA derivatives of IsoLG and MDA. IsoLG derivatives of 2-HOBA, using multiple reaction monitoring (MRM) of the m/z 456 → m/z 107 transition were reported in S.S. Davies et al, *Biochemistry* (2006) 45:15756-15767. This paper did not propose a structure for the m/z 107 product ion per se, but assumed it to be C₇H₇O⁺ by analogy to the m/z 152 product ion proposed for the IsoLG-adduct of pyridoxamine. No reactions with 4-HOBA were included in this manuscript. Subsequently, the reaction of MDA with both 2-HOBA and 4-HOBA was reported in I. Zagol-Ikapite et al, *J Lipid Res* (2015) 56:2196-2205. This paper used NMR to extensively characterize the structure of both the MDA-2-HOBA and the MDA-4-HOBA adducts. It also used mass spectrometry to characterize the product ions generated by fragmentation. Although the Supplementary Data of this *J Lipid Res* paper correctly reported that fragmentation of the MDA-2-HOBA [M+H]⁺ precursor ion gave rise to a product ion of m/z 107 ion, with a structure of C₇H₇O⁺, the main text of the paper erroneously reported the transition as being m/z 178 to m/z 106. In the original Suppl. Figure 6 for this current manuscript, the product

ion m/z 107 was erroneously drawn with the chemical formula C₇H₉N⁺. However, given that previous findings with 2-HOBA alone, with PITC derivatization of 2-HOBA, or with IsoLG modification of 2-HOBA had all also given rise to the m/z 107 product ion, and it seemed unlikely that fragmentation from the IsoLG-lactam adduct would retain the nitrogen atom as part of the product ion, we used high resolution mass spectrometry to identify the chemical formula of the m/z 107 ion for both 2-HOBA and 4-HOBA. For both 2-HOBA and 4-HOBA, the mass was m/z 107.049, consistent with the exact mass of 107.0491 calculated for C₇H₇O⁺ (and not with the mass of 107.073 that is calculated for C₇H₉N⁺). For both 2-HOBA

and 4-HOBA, the predicted product ion structures for C₇H₇O⁺ seem highly plausible as fragmentation ions of the original compounds.

Although this analysis confirmed that the same MRM transition can be used to monitor both 2-HOBA and 4-HOBA adducts, it is important to note that the retention times on HPLC differ slightly between these two regioisomers. For example, the PITC derivative of 2-HOBA has a retention time of 2.43 min while the PITC derivative of 4-HOBA has a retention time of 2.15 min in the chromatographic gradient used for these measurements (see the chromatographs in Suppl. Figure 16 panel A). The IsoLG adduct of 2-HOBA has a retention time of 3.08 min in the chromatographic gradient used for IsoLG-HOBA measurements, whereas the IsoLG adduct of 4-HOBA has a retention time of 2.99 min as shown in Suppl. Figure 16 panel B. Note that the MDA-2-HOBA and MDA-4-HOBA chromatographs have previously been published in I. Zagol-Ikapite et al, *J Lipid Res* (2015) 56:2196-2205.

One issue that became apparent from the high-resolution mass spectrometry analysis for the m/z 107 product ion was that the efficiency of formation of the m/z 107 product ion could potentially differ between 2-HOBA derivatives/adducts and the 4-HOBA derivatives/adducts. This can be seen in Suppl. Figure 16, where the ion intensity for PITC-2-HOBA is about 4-fold higher than that for the same amount of PITC-4-HOBA. (Conversely, the same amount of IsoLG-4-HOBA gives an approximately two-fold stronger signal than for the IsoLG-2-HOBA.)

In our studies where we measured 2-HOBA or 4-HOBA levels in tissues (after PITC derivatization), we use [²H₄]2-HOBA as our internal standard for both because we don't have deuterated 4-HOBA available to us currently. When we had originally calculated the plasma and tissue values for 4-HOBA, we had assumed that the concentration response for 4-HOBA was identical to that of 2-HOBA. Our finding that the product ion yield tended to be lower for 4-HOBA than for 2-HOBA led us to directly determine the concentration response of PITC-4-HOBA vs PITC-2-HOBA. For each sample, 1 nmol of [²H₄]2-HOBA was added as internal standard, and then 20-400 nmol of 4-HOBA or 2-HOBA added. The samples were derivatized with PITC and then measured using the following MRMs: for 2-HOBA or 4-HOBA m/z 259 → 107 and m/z → 153 (this second MRM represents the fragmentation of the PITC moiety), for [²H₄]2-HOBA m/z 263 → 111 and m/z 263 → 153. We then calculated the measured amount of 2-HOBA or 4-HOBA using the ratio of the peak area of the appropriate m/z 259 MRM to the peak area for the appropriate m/z 263 MRM. The results are now included as Suppl. Figure 17. While the concentration response for 2-HOBA using [²H₄]2-HOBA as the internal standard gave the expected slope of essentially 1, the concentration response for 4-HOBA using [²H₄]2-HOBA as the internal standard gave a slope of considerably less than 1. The ratio of the slopes for 2-HOBA/4-HOBA, when using the transition to the m/z 107 product ion was 3.9, while the ratio of the slopes when using the transition to the m/z 153 product ion was 5.7. Thus, these correction factors need to be applied to measurements of 4-HOBA.

We therefore reanalyzed our 30 min plasma samples from the oral gavage experiment with 5mg 2-HOBA or 4-HOBA of *Ldlr*^{-/-} mice fed WD for 16 weeks, using the m/z 107 product ion transition and applying a correction factor of 3.9 for the 4-HOBA samples (Suppl. Figure 4A). We also measured the levels of 2-HOBA or 4-HOBA in the aorta and heart of *Ldlr*^{-/-} mice fed chow diet 30 min after oral gavage (Suppl. Figures 4B and 4C). In addition, we performed a time course study after IP injection of 2-HOBA or 4-HOBA in C57BL6 mice that had been fed a standard chow diet to examine whether there were significant differences in the ADME characteristics of the two compounds. When this correction factor was applied, the amount of 4-HOBA seen in plasma 30 min after oral gavage was not statistically significantly different than the amount of 2-HOBA seen in plasma 30 min after oral gavage (Revised Suppl. Figure 4). In addition, the levels of 2-HOBA versus 4-HOBA were similar in the aorta and heart of *Ldlr*^{-/-} mice 30 min after oral gavage (Suppl. Figures 4B and 4C).

Pharmacokinetics of 4-HOBA.

We next compared the rate of elimination of 2-HOBA vs 4-HOBA in C57BL/6 mice fed a standard chow diet after IP injection of either 1 mg 2-HOBA or 1 mg 4-HOBA (Revised Suppl. Figure 5A). During the initial distribution phase, the maximal plasma concentration of 4-HOBA was somewhat higher than that of 2-HOBA (4-

HOBA, 22.9±4.3ug/mL; vs 2-HOBA, 13.9±0.9 ug/mL, mean±SEM). Thereafter, 4-HOBA appeared to undergo elimination more rapidly than 2-HOBA. Although the concentrations of 4-HOBA were significantly lower than that of 2-HOBA at 120 min and 240 min post-injection, it did not significantly differ at the other time points (applying t-test for each time point.) In addition, the levels of 2-HOBA versus 4-HOBA were similar in the spleen, liver, and kidney 30min after IP injection (Suppl. Figure 5B-5D).

Therefore, while there are differences in ADME characteristics in 4-HOBA and 2-HOBA and absolute concentrations of the two compounds do not perfectly match, we believe that given their overall similarity, it is reasonable to interpret the lack of 4-HOBA efficacy as being due to its lack of scavenging efficacy rather than simple due to lower concentrations.

Separate to this, the additional experiments (Suppl. Figure 8) do not establish the suitability of 4-HOBA as an ineffective dicarbonyl scavenger control. This is because analysis of “propenal-HOBA adduct” is based entirely on SRM of the transition m/z 178 \diamond 107. While this transition is reasonably assigned to the propenal adduct of 2-HOBA, none of the experiments reported establishes that it is also appropriate for the detection of the propenal adduct of 4-HOBA. Therefore, the absence of “propenal-HOBA adduct” after exposure of macrophages to oxLDL plus 4-HOBA does not establish a lack of reaction of 4-HOBA with the reactive bicarbonyl MDA in a biological system, and hence 4-HOBA as an inefficient dicarbonyl scavenger, just as the lack of in vivo protection by 4-HOBA cannot be taken as evidence for 2-HOBA mediating protection via dicarbonyl scavenging.

It is important to note that 4-HOBA will react with dicarbonyls including both MDA and IsoLG, but that for dicarbonyls like IsoLG or MDA, the rate of the reaction with 4-HOBA is ~100 times slower than the rate of reaction with 2-HOBA. All reactions to generate authentic standards for aldehyde-4-HOBA adducts were allowed to go to completion by incubating overnight. While the high reactivity of 2-HOBA allows it to outcompete proteins for reaction with dicarbonyls, the poor reactivity of 4-HOBA means that the only way it can compete is by being at much higher concentration.

As discussed above, 2-HOBA and 4-HOBA are positional isomers, so that their aldehyde/electrophile adducts share identical mass, but have slightly different retention times. We can detect 4-HOBA adducts during in vitro experiments if we incubate the 4-HOBA with the aldehyde for long periods of time. The lack of a detectable signal for MDA-4-HOBA or IsoLG-4-HOBA in 4-HOBA treated animals is not due to our inability to detect these adducts if they are present. The most plausible explanation for our failure to detect significant amounts of 4-HOBA adduct is that they are not generated in vivo in significant abundance. So, we assert that the fact that we can detect MDA-2-HOBA and IsoLG-2-HOBA adducts in 2-HOBA treated animals does in fact provide strong support for the notion that 2-HOBA acts as a dicarbonyl scavenger in vivo while 4-HOBA does not. Indeed, this is greatly substantiated by the finding that the urine of *Ldlr*^{-/-} mice collected during 16 h post oral gavage of 2-HOBA versus 4-HOBA contained 19-fold more MDA-HOBA adducts (Suppl. Figure 7A). Also, consistent with this concept is that the levels of MDA-HOBA adducts were markedly increased in the liver, kidney, and spleen 16 h post oral gavage after 2-HOBA treatment compared to 4-HOBA (Suppl. Figure 7).

Also, the additional studies do not justify the concentrations of HOBA used for in vitro experiments because the amount of HOBA administered by bolus (5 mg) corresponds to the amount of HOBA consumed over 24 h, with 1 g HOBA/L drinking water and assuming a consumption of 5 mL drinking water per day for a 30g mouse (Physiol Behav 2007;91:620). This discrepancy in administered dose is reflected in the plasma concentration of 2-HOBA determined 30 min (single time point) after bolus addition (~40,000 ± 10,000 ng/mL; Suppl. Figure 4A) being 2 orders of magnitude higher than the steady state concentration of 2-HOBA determined 16 weeks after treatment with 1 g 2-HOBA per litre drinking water, i.e., 469 ± 38 ng/mL. As a result, there remain fundamental problems with the proposed mode of anti-atherosclerotic activity of 2-HOBA.

It was not our intent to try to compare the bolus experiment with the 16-week treatment experiments. The bolus experiment was done to maximize formation of the 2-HOBAs or 4-HOBA adducts in order to identify the adducts

and their metabolites. In addition, the use of the oral bolus dose and the IP dose eliminate the possible variation in the dose administered due to variation in drinking habits of individual mice. The dose used in the 16-week studies (1 g/l) was based on previously published *in vivo* experiments when 2-HOBA was effective in preventing hypertension (A. Kirabo et al, *J Clin Invest.* [2014] 124:4642–465) and memory deficit (S. Davies et al, *J Alzheimers Dis.* [2011] 27:49–59) in mice. We recognize that because of the relatively short half-life of 2-HOBA in mice (62 min), and because mice tend to consume relatively little water during the day (since they are nocturnal animals), that there are likely to be significant diurnal variations in the levels of 2-HOBA, when it is administered in drinking water, and that the day time levels of 2-HOBA (when the mice were euthanized) potentially represents the nadir of 2-HOBA and 4-HOBA concentrations.

2. The concentration of HOBA used for *in vitro* experiments is ~1,000-fold higher than the steady state 2-HOBA concentration in plasma under conditions where anti-atherosclerotic activities are observed. Even compared with 2-HOBA plasma concentrations observed acutely after a bolus administration of a daily dose of 2-HOBA (that does not reflect the steady-state concentrations of the inhibitor), the *in vitro* concentrations used are 10-fold higher. Based on these findings, it is doubtful that the *in vitro* experiments meaningfully recapitulate the *in vivo* conditions.

While we think it is useful to consider the concentrations observed *in vivo* and the concentration used *in vitro*, we note that for many pharmaceutical compounds there are often substantial differences in concentrations of compounds used *in vitro* and what can be observed after dosing *in vivo*. Given that atherosclerosis is considered a chronic, smoldering inflammatory disease, while cell culture models of necessity use acute, high-intensity inflammatory stimulus, the need for higher concentrations of scavenger does not seem incongruous. For these reasons, we assert that the *in vitro* experiments provide robust evidence that 2-HOBA exerts scavenging effects even in a complex matrix, and, when added to our finding of MDA-2-HOBA and IsoLG-2-HOBA adducts *in vivo*, these results support the hypothesis that the beneficial effects of 2-HOBA *in vivo* most likely derive from scavenging of dicarbonyls. In addition, we examined the effects of a dose range of 2-HOBA on the expression of cytokines in macrophages in response to H₂O₂ and found that there were significant effects with 5 μM 2-HOBA (615 ng/mL) which is in the same range as the concentrations at “steady state” *in vivo* (Suppl. Figure 10). Importantly, we also found that during oxidative stress, 2-HOBA-MDA adducts were formed in cells during incubation with only 5 μM 2-HOBA (Suppl. Figure 11).

3. The urinary concentrations of F₂-IPs observed (Supplementary Figure 8) suggest that 2-HOBA or 4-HOBA have no effect on systemic oxidative stress. Unfortunately, the new data does not provide direct information on whether 2- and 4-HOBA comparably affect the PUFA-adjusted concentrations of F₂-IPs within lesions, the site where 2-HOBA is proposed to act as dicarbonyl scavenger.

This is an interesting suggestion, but the aortic samples from the atherosclerosis studies were completely used for the other analyses in our manuscript. However, we note that previous studies have demonstrated that F₂-IP in plasma and urine arise from esterified F₂-IP formed in tissues, and that urine F₂-IP levels correlate with extent of atherosclerotic lesion F₂-IP and with extent of atherosclerotic disease. Our goal was to examine whether 2-HOBA was behaving like an antioxidant and decreasing systemic oxidative stress. Previous studies have shown that treatment of apoE deficient mice with the antioxidant vitamin E reduced the extent of atherosclerosis and urine F₂-IP (D. Praticò et al. *Nat Med.* [1998] 4:1189-92). Therefore, we assert that the lack of changes in urine F₂-IP does in fact support the lack of a robust systemic anti-oxidant effect of 2-HOBA, including in atherosclerotic lesions. Lipid peroxidation generates many products including F₂ isoprostanes, isolevuglandins, and MDA. If 2-HOBA was acting to lower lipid peroxidation generally, it would reduce the levels of all of these. If instead, as we hypothesize, 2-HOBA acts as a dicarbonyl scavenger, we would expect that the levels of dicarbonyls like IsoLG and MDA and their adducts would be reduced, while non-dicarbonyl products of lipid peroxidation like F₂-isoprostanes would not change. Given that these are the results we see, this provides support for the hypothesis that 2-HOBA acts primarily via dicarbonyl scavenging.

4. It is appreciated that the authors have performed additional LC-MS analyses of lesion material to provide quantitatively more convincing evidence that 2-HOBA indeed acts as a dicarbonyl scavenger in atherosclerotic

lesion. Unfortunately, however, the new data provided (Figures 2C and 2D) lack data for vehicle, so that it is not possible from this data to conclude that 2-HOBA decreases MDA- and IsoLG-protein adducts. Comparing MDA- and IsoLG-protein adducts in 4-HOBA versus 2-HOBA does not overcome this deficiency. This is because such data is complicated to interpret because it is not clear whether the MS/MS transitions used, based on fragmentation of IsoLG- and MDA-2-HOBA, are identical and hence relevant for IsoLG- and MDA-4-HOBA (see also comment 1 above). The additional LC-MS/MS data provided in Suppl. Figure 5 add further concern. Specifically, it is unclear why the internal standard [²H₄]IsoLG-2-HOBA added to livers of 2-HOBA treated mice gives an m/z 476.3 ◊ 111.1 signal nearly 10-times higher than that seen with livers from 4-HOBA treated animals. In addition, the protein adducts reported in new Figure 2C and 2D should be standardised to lipid rather than protein, because IsoLG and MDA are derived from lipid oxidation and the lesion lipid content is different in 2-HOBA versus 4-HOBA-treated mice.

The reviewer's concerns about whether we have used the appropriate MRM transitions to monitor 4-HOBA adducts are addressed in our response to #1 above, so we will only respond to the additional concerns raised here. We are puzzled by the reviewer's statement that without a vehicle comparison it is not possible to conclude that 2-HOBA decreases MDA- and IsoLG-protein adducts. 4-HOBA and vehicle did not significantly differ in terms of disease, so the critical question that is being asked here is whether there was a significant difference in adduct levels between 2-HOBA treated animals and control groups with greater disease burden. A comparison to vehicle would not be as meaningful as a comparison to 4-HOBA, since 4-HOBA has a greater likelihood of mirroring any potential non-scavenging effects of 2-HOBA such as antioxidant effects. Our finding that there are significant differences between 2-HOBA and 4-HOBA are therefore more meaningful than if we simply found a difference between 2-HOBA and vehicle.

In terms of the question why in these particular studies the internal standard is ten-fold higher in the 2-HOBA treated animals than the 4-HOBA treated animals, we agree that a 10-fold difference in internal standard peak intensity is somewhat unusual, and a two- or three-fold variation is much more typical. Still, we note that the purpose of the internal standard is to account for any variability in extraction or other work-up procedure including variation of signal intensity on the mass spectrometry, and we do not have evidence that 4-HOBA adducts directly interfere with the [²H₄]IsoLG-2-HOBA signal, so we think it is most likely simply to be due to random variation.

The reviewer states that *protein* adducts should be normalized to PUFA rather than protein. The reviewer's desire for normalization against PUFA (presumably they specifically mean arachidonic acid since that is the only PUFA that gives rise to IsoLG, although the other PUFA can give rise to MDA) appears to stem from the reviewer's concern that 2-HOBA could potentially reduce IsoLG and MDA adduct levels in the aortic lesions by some indirect effect, independent of direct scavenging. Primarily the reviewer appears to be concerned that 2-HOBA could act by some non-scavenging dependent mechanism to reduce lipid burden in the plaque, so that the reduced levels of MDA- or IsoLG- adducts are merely the byproduct of reduced PUFA levels in the lesion. For the normalization to PUFA or arachidonic acid to be more valid than normalization to protein, it would need to be the case that the amount of PUFA or arachidonic acid/mg protein or the amount of PUFA or arachidonic acid per mg of wet tissue weight is higher in atherosclerotic aorta than in healthy/control aorta. However, we note that as far as we can tell, the studies that have looked at arachidonic acid or PUFA levels in lesion vs control tissue do not support the notion that this is the case. Many years ago, Toussaint et al³ used NMR to study PUFA levels in human arteries, comparing arteries with low levels of obstruction (<40%) to high levels of obstruction (>40%). They found reduced NMR signals for PUFA in arteries with high obstruction. More recently, Szklenar et al⁴ performed lipidomic studies comparing the levels of a large number of eicosanoids in male New Zealand White rabbits fed a high cholesterol diet to induce atherosclerosis vs rabbits fed control diet. This rabbit model is widely used to study atherosclerosis. This study found no significant difference in the levels of arachidonic acid or linoleic acid per g aorta in the high cholesterol fed rabbits compared to controls. Some, but not all, species of oxygenated PUFA varied between control and atherosclerotic aortas, suggesting that the activity of oxidative enzymes rather than the burden of PUFA was the most important factor in these differences. There were also no significant changes in plasma PUFA levels (when normalized per ml of plasma) found in this study. We have not been able to find studies in the literature reporting arachidonic acid or PUFA levels in the aorta of atherosclerotic mice compared to controls, but we would anticipate they would follow what is seen in humans and rabbits.

Therefore, we do not believe that normalization to arachidonic acid or PUFA is more appropriate than normalization to protein or that this normalization would be likely to significantly change the interpretation of our data. Furthermore, we do not have any aortic tissue left from the atherosclerosis studies as it was all used for the analyses reported in this manuscript, so we believe these experiments are beyond the scope of the current manuscript.

5. It is appreciated that carrying out intervention studies is beyond the scope of the present investigation, so that stating the limitation of the present study is appropriate. Notwithstanding this, however, the authors' view that "the prevention of formation of atherosclerotic lesion is an important strategy for the prevention of cardiovascular disease" can be challenged. I am not aware of any drug-related strategy currently in use that is commenced before atherosclerotic lesions are present, and such strategy seems unlikely to be relevant in the near future, not least because treatment would have to commence in childhood and continue throughout life. Is this really what the authors have in mind for 2-HOBA, or am I missing something?

Whether you are implementing either primary prevention (no known ASCVD) or secondary prevention (known ASCVD) strategies with lipid lowering therapy, the goal is not only to treat any preexisting lesions but also to prevent the formation of new atherosclerotic lesions. It is standard of care to treat children with heterozygous FH with lipid lowering therapy as early as age 8 with the goal of not only treating any atherosclerotic lesions that might exist but also to prevent the formation of new atherosclerotic lesions. Based on the regression studies, we know that statin therapy promotes relatively small changes in preexisting atherosclerotic lesions but exerts large clinical benefits. Early atherosclerotic lesions are more likely to undergo regression than advanced complex lesions. In studies of experimental atherosclerosis, it is widely accepted to study the ability of interventions to prevent the formation of atherosclerotic lesions. At this point, we are not advocating the use of 2-HOBA clinically in any specific context or time frame, because it has not been studied for this purpose in clinical trials. However, there should not be any reason not to point out that a drug, being used to prevent experimental atherosclerosis, is successful in doing so, or to suggest that the ability to prevent atherosclerotic lesion formation is anything other than a highly desirable property.

6. The point raised here is two-fold: First, dicarbonyls are derived from PUFA (and perhaps more precisely bisallylic hydrogen-containing lipids) so that the dicarbonyl content is likely affected by the PUFA content. Second, if 2-HOBA decreases lesion size, it is expected to decrease the pool of PUFA from which dicarbonyls are formed, so that adduct formation needs to be lipid standardised (see comment 4).

This issue of normalization to lipid is addressed in our response to comment 4 above.

7. OK

Minor Points:

8. OK

9. OK

10. OK

11. OK

12. OK.

13. OK

14. OK

15. Non-parametric or parametric tests were used for statistical analyses of the data. Please justify why a specific statistical method was used (performing normality and equal variance tests for continuous variables).

Continuous data are summarized as mean \pm SEM visualized by box plots and bar charts. Between-group differences were assessed with Student's t-test (2 groups) and one-way ANOVA (> 2 groups, Bonferroni's correction for multiple comparisons). Their nonparametric counterparts, Mann-Whitney test (2 groups) and nonparametric Kruskal-Wallis test (more than 2 groups, Bunn's correction for multiple comparison) were used when assumptions for parametric methods were not met. The Shapiro-Wilk-Wilk test was used to evaluate normality assumptions. All tests were considered statistically significance at two-sided significance level of 0.05

after correction for multiple comparisons. All statistical analyses were performed in GraphPad PRISM versions 5 or 7. In addition, relevant information regarding statistics is included in the Figure Legends.

Reviewer #3: (Remarks to the Author):

In the revised version, the authors added novel results which have satisfied my concerns. In particular, the identification of adducts between 2-HOBA and di-aldehydes strengthens the carbonyl sequestering mechanism of the therapeutic agent, further sustained by the F2-isoprostanes results. In my opinion, the paper is of great interest because it reveals novel and important aspects of carbonylation stress and in particular identify di-carbonyls as pathogenetic factors of atherosclerosis. Moreover, the paper furnishes convincing evidences of a potential drug target and of a new class of pharmacological tools. The present paper will lead to a further great scientific interest to carbonyl stress and to sequestering agents able to prevent it.

We greatly appreciate the assessment from Reviewer 3 that the identification of adducts between 2-HOBA and di-aldehydes strengthens the carbonyl sequestering mechanism of the therapeutic agent, further sustained by the F2-isoprostanes results.” We hope that Reviewer 3 will agree that our revised manuscript has been further strengthened.

Reviewers' Comments:

Reviewer #1:

Remarks to the Author:

none

Reviewer #4:

Remarks to the Author:

The relative effect of 2-HOBA vs 4-HOBA on atherogenesis is unequivocal; many interventions limit atherogenesis and fewer reverse or even restrain established lesions from further progression. It is assumed that these last two augment the translational relevance of the intervention.

The adjusted calculations of relative HOBA availability largely address the concerns originally expressed although some difference in systemic availability persists and the use of a single deuterated standard for the two compounds is a potential source of error. The evidence for restriction to the dicarbonyls afforded by lipid peroxidation measurements is acceptable.

The rationale for use of much higher concentrations used in the in vitro experiments than is achieved in vivo is highly arguable.

The human FH data are a marginal contribution to a paper which largely revolves around (i) the use of a pharmacological probe in a mouse model of atherogenesis (ii) the acceptance of 4-HOBA administration and the measures of lipid peroxidation to support the argument for specificity of its effect.

Response to the Reviewers' Comments:

We are gratified that R1 was satisfied with our responses and had no further comments. We appreciate R4's insightful comments and constructive criticism. We have responded in detail to R4's comments, and we believe that addressing these comments has significantly improved our manuscript. We hope that you will agree that we have adequately addressed the issues that were raised.

Reviewer #4: (Remarks to the Author):

1) The relative effect of 2-HOBA vs 4-HOBA on atherogenesis is unequivocal; many interventions limit atherogenesis and fewer reverse or even restrain established lesions from further progression. It is assumed that these last two augment the translational relevance of the intervention.

We agree that the atherosclerosis results are unequivocal and that our results support the translational relevance of the intervention of dicarbonyl scavengers as a potential treatment for atherosclerosis.

2) The adjusted calculations of relative HOBA availability largely address the concerns originally expressed although some difference in systemic availability persists and the use of a single deuterated standard for the two compounds is a potential source of error. The evidence for restriction to the dicarbonyls afforded by lipid peroxidation measurements is acceptable.

“We are grateful that R4 felt that we had largely addressed the concerns about bioavailability. In response to R4's comments, we have also added text to the methods section acknowledging the limitation of our quantitative approach for 4-HOBA, this reads as follows:

Page 14: One limitation of using [²H₄]-2-HOBA as an internal standard to measure 4-HOBA is that it required the use of an external calibration curve to calculate a correction factor. This method is not as accurate, but deuterated 4-HOBA was not available.

3) The rationale for use of much higher concentrations used in the in vitro experiments than is achieved in vivo is highly arguable.

We now provide our rationale for using the higher concentrations of 2-HOBA in the in vitro experiments and recognize it as a limitation by the following statement.

Page 10: “An important limitation of these in vitro studies is that a relatively high concentration of 2-HOBA (500 uM) was used for most of these experiments, in part because we used a high concentration of H₂O₂ (250 uM) to maximize the inflammatory response and apoptosis induced in these studies. While similar 2-HOBA concentrations can be achieved immediately following gavage of the entire daily intake of 2-HOBA (5 mg, as in Supplemental Figure 4), steady state concentrations are much lower (i.e. 3-5 uM 2-HOBA) when the ~5 mg dose is administered by drinking water throughout the day as was done in our primary in vivo studies²². However, it is important to note our finding that 5 uM 2-HOBA sufficed to significantly reduce macrophage inflammatory cytokine expression in macrophages when a lower concentration of H₂O₂ (100 uM) was applied (Supplementary Figure 10).”

4) The human FH data are a marginal contribution to a paper which largely revolves around (i) the use of a pharmacological probe in a mouse model of atherogenesis (ii) the acceptance of 4-HOBA administration and the measures of lipid peroxidation to support the argument for specificity of its effect.

The reviewer's points are well taken. Although, we appreciate the fact that most of our data relates to the impact of 2-HOBA on lipoprotein modification and atherosclerosis in *Ldlr*^{-/-} mice, with a relatively small contribution of data showing that HDL is modified and dysfunctional in humans with FH, we believe that the human FH data plays an important role in substantiating therapeutic targets and highlighting the potential translational relevance of dicarbonyl scavenging as a therapeutic approach to reduce lipoprotein modification and improve HDL function in humans with FH. Despite tremendous advances in developing medications that lower LDL-C and reduce cardiovascular events, patients with FH continue to have high residual risk of cardiovascular events. Furthermore, individuals with FH are underdiagnosed and undertreated. Obviously, there are limitations to the extent to which the potential benefits and safety of pharmacological treatments in animals can be extrapolated to humans, but these studies are done with the ultimate hope that the therapy can be translated into humans. Given that 2-HOBA has been shown to be safe in two Phase I trials in humans (refs), we believe the current results provide strong support for performing a translational phase II randomized trial of the impact of 2-HOBA on lipoprotein modification and function in humans with FH. We have now provided the following statement on page 10.

“Although, *Ldlr*^{-/-} mice represent a relevant murine model of FH, there are limitations to the extent that our results from treatment of *Ldlr*^{-/-} mice with 2-HOBA can be extrapolated to the anticipated therapeutic results in humans with FH or coronary artery disease. However, given that the initial phase I studies have demonstrated the safety of 2-HOBA in humans^{1, 2}, we believe that our results support the importance of future translational studies to evaluate the impact of 2-HOBA on lipoprotein modification and function in humans with FH.”

1. Pitchford LM, *et al.* First-in-human study assessing safety, tolerability, and pharmacokinetics of 2-hydroxybenzylamine acetate, a selective dicarbonyl electrophile scavenger, in healthy volunteers. *BMC Pharmacol Toxicol* **20**, 1 (2019).
2. Pitchford LM, *et al.* Safety, tolerability, and pharmacokinetics of repeated oral doses of 2-hydroxybenzylamine acetate in healthy volunteers: a double-blind, randomized, placebo-controlled clinical trial. *BMC Pharmacol Toxicol* **21**, 3 (2020).